



# Inverting ice surface elevation and velocity for bed topography and slipperiness beneath Thwaites Glacier

Helen Ockenden[1], Robert G. Bingham[1], Andrew Curtis[1], and Daniel Goldberg[1]

[1]School of GeoSciences, University of Edinburgh, Drummond St, Edinburgh EH8 9XP

**Correspondence:** Helen Ockenden (h.ockenden@sms.ed.ac.uk)

**Abstract.** There is significant uncertainty over how ice sheets and glaciers will respond to rising global temperatures. Limited knowledge of the topography and rheology of ice-bed interface is a key cause of this uncertainty, as models show that small changes in the bed can have a large influence on predicted rates of ice loss. Most of our detailed knowledge of bed topography comes from airborne and ground-penetrating radar observations. However, these direct observations are not spaced closely

enough to meet the requirements of ice-sheet models, so interpolation and inversion methods are used to fill in the gaps. Here we present the results of a new inversion of surface-elevation and velocity data over Thwaites Glacier, West Antarctica, for bed topography and slipperiness (i.e. the degree of basal slip for a given level of drag). The inversion is based on a steady-state linear perturbation analysis of the shallow-ice-stream equations. The method works by identifying disturbances to surface flow which are caused by obstacles or sticky patches in the bed, and can therefore be applied wherever the shallow-ice-

stream equations hold and where surface data are available, even where the ice thickness is not well known. We assess the performance of the inversion for topography with the available radar data. Although the topographic output from the inversion is less successful where the bed slopes steeply, it compares well with radar data from the central trunk of the glacier. This method could therefore be useful as either an independent test of other interpolation methods such as mass conservation and kriging, or as a complementary technique in regions where those techniques fail. We do not have data to allow us to assess the

success of the slipperiness results from our inversions, but we provide maps that may guide future seismic data collection across Thwaites Glacier. The methods presented here show significant promise for using high-resolution satellite datasets, calibrated by the sparser field datasets, to generate high resolution bed topography products across the ice sheets, and therefore contribute to reduced uncertainty in predictions of future sea-level rise.

## 1 Introduction

Predicting the rate at which marine sectors of the West Antarctic Ice sheet will retreat and contribute to globally rising sea levels is of increasing importance due to persistent climate forcing across the region over the last decades (Scambos et al., 2017; Turner et al., 2017). Ice-sheet modelling studies emphasise the role of bed topography and rheology in understanding





future ice loss (Durand et al., 2011; Parizek et al., 2013; Sun et al., 2014; Nias et al., 2016, 2018; Kyrke-Smith et al., 2018; Yu
et al., 2018; Koellner et al., 2019). Bed topography is particularly important for marine-terminating glaciers, such as Thwaites
Glacier in West Antarctica, which are vulnerable to the marine ice-sheet instability (Weertman, 1974; Hughes, 1981; Schoof,
2007; Goldberg et al., 2009; Gudmundsson, 2013). However, bed topography constrained by geophysical surveying at the
resolutions required for ice-sheet modelling (Durand et al., 2011; McCormack et al., 2018) is rarely available, so projections of
future ice-sheet behaviour have to rely on bed topographies interpolated in a variety of ways between the direct measurements
(Vaughan et al., 2006; Fretwell et al., 2013; Rignot et al., 2014; Millan et al., 2017; Morlighem et al., 2020). Over Thwaites
Glacier, these interpolations have typically infilled areas of 15 by 15 km between aerogeophysical flight lines, but 15 km
between observations is much coarser than the resolution which Durand et al. (2011) suggest is desirable.

Where ice-penetrating radar surveys have been undertaken with sub-ice-thickness line spacing (eg. Rutford Ice Stream - King
et al., 2016; Pine Island Glacier - Bingham et al., 2017; Thwaites Glacier - Holschuh et al., 2020), they clearly identify details
which are important for studying future ice-sheet behaviour that are not present in the interpolated bed topography products.
In particular, imaged signatures in the bed often show some similarity to the much subtler topography of the ice surface above
them. Theoretical studies based on linear perturbation theory (Gudmundsson, 2003; Raymond, 2005; Gudmundsson, 2008;
Gudmundsson and Raymond, 2008; Raymond and Gudmundsson, 2009) have explored the relationship between the bed and
the surface. The resulting relations can be used to infer bed properties from those of the surface, but have only been applied
twice to realistic settings: on 2D surface data from MacAyeal Ice Stream (Thorsteinsson et al., 2003), and on a 1D flow line
from Rutford Ice Stream (Pralong and Gudmundsson, 2011). Both studies were undertaken in an era when surface elevation
observations over Antarctica were of much lower quality and resolution than they are today.

Bed conditions such as geology, hydrology and sediment distribution also play a role in controlling ice flow and behaviour
(Durand et al., 2011; Koellner et al., 2019), and are often poorly constrained. In many ice-sheet models, these bed conditions
are combined into one parameter known as slipperiness, which is a measure of how easily the ice can slide over the topog-
raphy (Rignot et al., 2011). Some seismic lines have been collected on Thwaites Glacier (Muto et al., 2019a, b), allowing a
brief glimpse into the sediment distribution. Over the whole glacier, however, there are very few direct measurements of bed
conditions which can be combined into slipperiness.

In this paper we exploit the relatively new availability of high resolution surface elevation (REMA, ∼ 8 m, Howat et al.,
2019) and velocity (NASA ITS_LIVE, ∼ 120 m resolution, Gardner et al., 2018) datasets. We apply the under-used linear-
perturbation theories to explore bed topography and slipperiness across the Thwaites Glacier catchment. We use a steady-state
version of the shallow-ice-stream equations presented by Gudmundsson (2008) and compare the topography output from the
inversion to radar grids and flight lines to assess its performance.



## 2 Methodology

### 2.1 Derivation of the steady-state shallow-ice stream transfer functions

Gudmundsson (2008) derived a set of transfer functions which describe the relationship between the time-variant Fourier transforms of bed topography ($\hat{b}$), bed slipperiness ($\hat{c}$), surface topography ($\hat{s}$) and horizontal components of surface velocity ($\hat{u}, \hat{v}$). For the purposes considered here, this derivation can be simplified by considering the steady state from the beginning, removing the need to do a Laplace transform. Other than this, the derivation largely follows that of Gudmundsson (2008), but for clarity we state key assumptions and results here.

#### 2.1.1 Response of flow to basal topography perturbations

We start with the shallow-ice-stream equations of motion (MacAyeal, 1989),

$$\partial_x(4h\eta\partial_x u + 2h\eta\partial_y v) + \partial_y(h\eta(\partial_x v + \partial_y u)) - (u/c)^{1/m} = \rho gh\partial_x(s)\cos\alpha - \rho gh\sin\alpha \tag{1}$$

$$\partial_y(4h\eta\partial_y v + 2h\eta\partial_x u) + \partial_x(h\eta(\partial_y u + \partial_x v)) - (v/c)^{1/m} = \rho gh\partial_y(s)\cos\alpha \tag{2}$$

where $u$, $v$, $w$ are the velocity components in the $x, y$ and $z$ directions respectively, $h$ is the ice thickness, $\eta$ is the effective ice viscosity, $c$ is the basal slipperiness, $m$ is a sliding law parameter, $\rho$ is the ice density, $g$ is the acceleration due to gravity, $s$ is the ice surface elevation, $b$ is the ice bed elevation, and $\alpha$ is the mean ice surface slope in the $x$ direction.

We consider the response to a small perturbation in basal topography, $b$, linearising around a reference model $(\bar{h}, \bar{s}, \bar{b}, \bar{u}, \bar{v}, \bar{c})$ with $h = \bar{h} + \Delta h$, $s = \bar{s} + \Delta s$, $b = \bar{b} + \Delta b$, $u = \bar{u} + \Delta u$, $v = \Delta v$, $w = \Delta w$ and $c = \bar{c}$.

This gives the first order momentum balance equations

$$4\eta\bar{h}\partial_{xx}^2\Delta u + 3\eta\bar{h}\partial_{xy}^2\Delta v + \eta\bar{h}\partial_{yy}^2\Delta u - \gamma\Delta u = \rho g\bar{h}\cos\alpha\partial_x\Delta s - \rho g\sin\alpha\Delta h \tag{3}$$

$$4\eta\bar{h}\partial_{yy}^2\Delta v + 3\eta\bar{h}\partial_{xy}^2\Delta u + \eta\bar{h}\partial_{xx}^2\Delta v - \gamma\Delta v = \rho g\bar{h}\cos\alpha\partial_y\Delta s \tag{4}$$

Also to the first order, and importantly in the steady state, we have the upper and lower kinematic boundary conditions

$$\bar{u}\partial_x\Delta s - \Delta w(s) = 0 \tag{5}$$

$$\bar{u}\partial_x\Delta b - \Delta w(b) = 0 \tag{6}$$

All variables are then Fourier transformed with respect to the spatial variables $x$ and $y$. In the forward Fourier transform the wavenumbers in the $x$ and $y$ directions are denoted by $k$ and $l$ respectively. This Fourier transform gives

$$4\eta\bar{h}k^2\hat{u} + 3\eta\bar{h}kl\hat{v} + \eta\bar{h}l^2\hat{u} + \gamma\hat{u} = \rho g\bar{h}\cos\alpha ik\hat{s} + \rho g\sin\alpha\hat{h} \tag{7}$$

$$4\eta\bar{h}l^2\hat{v} + 3\eta\bar{h}kl\hat{u} + \eta\bar{h}k^2\hat{v} + \gamma\hat{v} = \rho g\bar{h}\cos\alpha il\hat{s} \tag{8}$$

$$\hat{w}(\bar{s}) = -i\bar{u}k\hat{s} \tag{9}$$

$$\hat{w}(\bar{b}) = -i\bar{u}k\hat{b} \tag{10}$$



where $\hat{h} = \hat{s} - \hat{b}$.

From depth integration of the Fourier-transformed incompressibility condition $w_z + u_x + v_y = 0$ we have

$$i\bar{h}(k\hat{u} + l\hat{v}) = \hat{w}(\bar{s}) - \hat{w}(\bar{b}) \tag{11}$$

which, along with the steady-state boundary conditions, yields

$$i\bar{h}(k\hat{u} + l\hat{v}) = -ik\bar{u}\hat{s} + ik\bar{u}\hat{b}. \tag{12}$$

Equations A7, A8 and A12 form a linear system of equations in $\hat{s}$, $\hat{u}$, $\hat{v}$ and $\hat{b}$ which can be solved algebraically (see Appendix A), leading to the steady-state transfer functions

$$T_{sb}(k,l) = \frac{\hat{s}}{\hat{b}} = \frac{ik(\bar{u}\xi + \tau_d)}{p\xi} \tag{13}$$

$$T_{ub}(k,l) = \frac{\hat{u}}{\hat{b}} = \frac{\tau_d \cot\alpha (l^2\tau_d - k^2\bar{u})}{\xi\nu p} \tag{14}$$

$$T_{vb}(k,l) = \frac{\hat{v}}{\hat{b}} = \frac{kl\tau_d\cot\alpha(\tau_d + \nu\bar{u})}{\xi\nu p} \tag{15}$$

which represent the ratio of variability in the Fourier components of the surface to variability in the Fourier components of the bed. The following abbreviations are used for simplicity in the derivation: $\xi = \gamma + 4\bar{h}j^2\eta$, $\quad \gamma = \frac{\tau_d^{1-m}}{m\bar{c}}$, $\quad j^2 = k^2 + l^2$,

$\tau_d = \rho g \bar{h} \sin\alpha$, $\quad p = \frac{i}{t_p} - \frac{1}{t_r}$, $\quad \frac{1}{t_p} = k\left(\bar{u} + \frac{\tau_d}{\xi}\right)$, $\quad \frac{1}{t_r} = \frac{j^2\tau_d\bar{h}\cot\alpha}{\xi}$, $\quad$ and $\nu = \gamma + \bar{h}j^2\eta$.

### 2.1.2 Response of flow to basal slipperiness perturbation

Starting once again with the shallow-ice-stream equations (Equations A1 and A2; MacAyeal, 1989), this time we consider the response to a small perturbation in basal slipperiness, $c$, linearising with $h = \bar{h} + \Delta s$, $s = \bar{s} + \Delta s$, $b = \bar{b}$, $u = \bar{u} + \Delta u$, $v = \Delta v$, $w = \Delta w$ and $c = \bar{c}(1 + \Delta c)$ where $\Delta c$ is the fractional slipperiness.

This gives the first order momentum balance equations

$$4\eta\bar{h}\partial_{xx}^2\Delta u + 3\eta\bar{h}\partial_{xy}^2\Delta v + \eta\bar{h}\partial_{yy}^2\Delta u - \gamma\Delta u = \rho g\bar{h}\cos\alpha\partial_x\Delta s - \rho g\sin\alpha\Delta s - \gamma\bar{u}\Delta c \tag{16}$$

$$4\eta\bar{h}\partial_{yy}^2\Delta v + 3\eta\bar{h}\partial_{xy}^2\Delta u + \eta\bar{h}\partial_{xx}^2\Delta v - \gamma\Delta v = \rho g\bar{h}\cos\alpha\partial_y\Delta s \tag{17}$$

Fourier transforming with respect to the spatial variables $x$ and $y$ gives:

$$4\eta\bar{h}k^2\hat{u} + 3\eta\bar{h}kl\hat{v} + \eta\bar{h}l^2\hat{u} + \gamma\hat{u} = \rho g\bar{h}\cos\alpha ik\hat{s} + \rho g\sin\alpha\hat{s} + \gamma\bar{u}\hat{c} \tag{18}$$

$$4\eta\bar{h}l^2\hat{v} + 3\eta\bar{h}kl\hat{u} + \eta\bar{h}k^2\hat{v} + \gamma\hat{v} = \rho g\bar{h}\cos\alpha il\hat{s} \tag{19}$$

As there is no bed topography perturbation, the steady-state boundary conditions become

$$i\bar{h}(k\hat{u} + l\hat{v}) = -ik\bar{u}\hat{s}. \tag{20}$$





Equations B3, B4 and B5 form a linear system of equations which can be solved using standard algebraic techniques (see Appendix B), leading to the steady-state transfer functions

$$T_{sc}(k,l) = \frac{\hat{s}}{\hat{c}} = -\frac{ik\bar{h}\bar{u}\gamma}{p\xi} \tag{21}$$

$$T_{uc}(k,l) = \frac{\hat{u}}{\hat{c}} = \frac{\gamma\bar{u}\Big((3\eta\bar{h}l^2+\nu)(ik\bar{u}) - l^2\tau_d\cot\alpha\bar{h}\Big)}{\xi\nu p} \tag{22}$$

$$T_{vc}(k,l) = \frac{\hat{v}}{\hat{c}} = \frac{kl\gamma\bar{u}\bar{h}\big(\tau_d\cot\alpha - 3i\eta\bar{u}k\big)}{\xi\nu p} \tag{23}$$

which represent the ratio of variability in the Fourier components of the surface to variability in the Fourier components of the slipperiness.

Note that the transfer functions $T_{uc}$ and $T_{vc}$ are not the same as the steady-state versions of the transfer functions published in Gudmundsson (2008), as there is a typographic error in their paper. However, when plotted graphically, they can be used to reproduce the figures in that paper.

### 2.1.3 Non-dimensionalisation

The form of these transfer functions can be simplified by considering them in a non-dimensional form. For this purpose the same scalings as used in Gudmundsson (2003) and Gudmundsson (2008) are employed. All spatial scales are in units of mean ice thickness ($\bar{h}$), and stress components are in units of driving stress ($\tau_d$). Non-dimensional velocity components are in units of mean deformational velocity ($\bar{u}_d$) where

$$\bar{u}_d = \frac{\bar{h}\tau_d}{2\eta}. \tag{24}$$

The scale for slipperiness is given by $\bar{c}/\bar{C}$. From Gudmundsson (2008) we know that $\bar{c}/\bar{C} = \bar{u}_d/\tau_d^m$ and we also have $\bar{C} = \bar{u}_b/\bar{u}_d$. The ice surface velocity is the sum of the deformational velocity and the basal velocity, such that $\bar{u}_s = \bar{u}_d + \bar{u}_b$. With some simple algebra, we can therefore express the scale for slipperiness in terms of the surface velocity $\bar{u}_s$, which is a known quantity,

$$\frac{\bar{c}}{\bar{C}} = \frac{\bar{u}_s}{\tau_d^m(\bar{C}+1)}. \tag{25}$$

The non-dimensional form of the equations is obtained using the substitutions $\bar{c} \mapsto \bar{C}, \eta \mapsto 1/2, \bar{h} \mapsto 1, \bar{u} \mapsto \bar{C}, \gamma \mapsto (m\bar{C})^{-1}$, and $\tau_d = \rho g\bar{h}\sin\alpha \mapsto 1$. The non-dimensional transfer functions can be found in the supplementary information, and are also shown graphically there. Non-dimensionalised parameters are represented by capital letters (B, C, S, U, V, $T_{SB}$, $T_{UB}$, $T_{VB}$, $T_{SC}$, $T_{UC}$ and $T_{UV}$).

### 2.2 The inverse problem

The non-dimensional transfer functions ($T_{SB}, T_{UB}, T_{VB}, T_{SC}, T_{UC}$ and $T_{VC}$) describe the relationship between the Fourier transforms of non-dimensionalised bed topography ($\hat{B}$), bed slipperiness ($\hat{C}$), surface topography ($\hat{S}$) and surface velocity



$(\hat{U}, \hat{V})$. If the bed topography and slipperiness are known then surface topography and velocity components are given by the forward model:

$$\hat{S} = T_{SB}\hat{B} + T_{sc}\hat{C} \tag{26}$$

$$\hat{U} = T_{UB}\hat{B} + T_{uc}\hat{C} \tag{27}$$

$$\hat{V} = T_{VB}\hat{B} + T_{vc}\hat{C} \tag{28}$$

Since the system is over-determined, we can solve for non-dimensional bed topography and slipperiness using a weighted least-squares inversion of equations 26, 27 and 28, with a filter applied to remove problematic wavelength components. Short-wavelength features and features aligned with ice flow are problematic because they cause flow disturbances in the ice which do not reach the surface in a measurable way, and so they can not be inverted from the surface data. Increasing the filtering parameter ($p_{filt}$) removes progressively longer wavelength features. This method was first used by Thorsteinsson et al. (2003) 
in their study of MacAyeal Ice Stream (formerly Ice Stream E). The equations which solve Equation 26, 27, 28 are therefore not repeated here in the main text, but are given in notation consistent with this paper in Appendix C.

### 2.3  Synthetic tests

Synthetic tests allow us to explore which bed features can and can not be resolved using this inversion method. First we create a synthetic bed topography ($b$) and slipperiness ($c$) on a 50 km by 50 km grid, with a 120 m data spacing, purposely 
matching the data spacing of the ITSLIVE velocity product (Gardner et al., 2018). We subtract the mean bed elevation, slope and slipperiness so that the bed varies about 0, and taper the outer 5 km of the grid linearly to 0 at the edges. This reduces edge effects in the inversion, because the Fourier transform requires a periodic domain. We tested a few other sensible tapering functions, including semi-sinusoids, but observed negligible differences in the inversion output when compared to the linear function. We then non-dimensionalise the tapered bed using the length scales given in Section 2.1.3, and Fourier transform 
to get $\hat{B}$ and $\hat{C}$. The non-dimensional surface elevation ($\hat{S}$) and the velocity components ($\hat{U}$, $\hat{V}$) are calculated using the forward model, and dimensionalised using the length scales given in Section 2.1.3. To simulate measurement errors in the real surface data, we add random noise to the generated surface ($s$, $u$, $v$). This noise is white noise with a Gaussian low pass filter applied in Fourier space to give it a non-random frequency distribution. We then taper, non-dimensionalise and Fourier transform the noise-added surface data. Finally, we invert for non-dimensional bed topography ($\hat{B}$) and slipperiness ($\hat{C}$) using 
the inversion procedure described in Section 2.2 and the supplementary information. After dimensionalisation, the inverted bed can be compared to the synthetic bed to study the behaviour of the inversion.

### 2.3.1  Parameter value choices

When running synthetic tests, several model parameters can be varied, in addition to the synthetic bed topography and slipperiness. Following Gudmundsson (2008) and Thorsteinsson et al. (2003) the sliding law constant was set to $m = 1$, and the 
filtering parameter (Equation C4) to $p_{filt} = -2$. The mean ice thickness $\bar{h}$, mean surface slope $\alpha$ and mean ice surface velocity





$\bar{u}$ depend on the region studied, but for these synthetic tests are set at $\bar{h} = 2000$ m, $\alpha = 0.002$, $\bar{u} = 100\text{ms}^{-1}$, values thought to be appropriate for the Thwaites Glacier region (Gudmundsson, 2008; Howat et al., 2019; Morlighem et al., 2020; Gardner et al., 2018). After studying the results of the Thwaites Glacier inversion for a variety of values of $\bar{C}$, the non-dimensional mean slipperiness, we set to be $\bar{C} = 100$. When applied to Equation 25, these values give a dimensional mean slipperiness

$\bar{c} = 2.7 \text{ x}10^{-3} \text{ myr}^{-1}\text{Pa}^{-1}$. We set the weighting factors (Equation C2) to be $\Sigma_s = 0.001$, $\Sigma_u = 1$ and $\Sigma_v = 1$. This accounts for the mismatch in the relative magnitude of the non-dimensionalised surface elevation and velocity, and means the least-squares inversion solves for all three factors equally.

### 2.3.2   Resolution of bed forms

A two-dimensional Fourier transform decomposes an image into a weighted sum of two-dimensional sinusoidal basis functions.

For this reason, all of our synthetic tests used sinusoidal bed topographies and slipperiness, as these are the most illustrative of the capabilities of the inversion. Sinusoidal basis functions vary depending on three parameter: the wavenumbers in the $x$ and $y$ directions $(k,l)$ and the weighting or amplitude of the sinusoid. However, rather than considering wavenumbers, it is more intuitive to consider the horizontal wavelength, $\lambda$, and angle, $\theta$, to the direction of the flow, where $j^2 = l^2 + k^2$, $\lambda/\bar{h} = 2\pi/j$, $k = j\cos(\theta)$, and $l = j\sin(\theta)$.

Figures 1, 2 and 3 show how well bed topography and slipperiness can be resolved by the inversion when angle to flow, $\theta$, wavelength, $\lambda$, and amplitude are varied. The amplitude of the noise added to the synthetically generated surface is $\pm 2$ m for surface elevation and $\pm 15$ m s$^{-1}$ for velocity components as these are at the upper limit of the errors for REMA (Howat et al., 2019) and ITSLIVE (Gardner et al., 2018) in the Thwaites Glacier region.

These synthetic tests show that in this simple least-squares inversion, the bed can be well resolved if the angle to the flow

is greater than 15 degrees, the wavelength is more than 2000 m and the amplitude is greater than 10 m for topography or $1.34 \text{ x}10^{-4} \text{ myr}^{-1}\text{Pa}^{-1}$ for variability in slipperiness. It is worth noting, however, that the resolution of wavelengths varies depending on the ice thickness, which is the non-dimensional scale factor for lengths. This means that ice thickness is directly proportional to the wavelength at which variations in bed topography and slipperiness should be resolvable.

### 2.4   Applying the inversion to real data

We now turn our attention to the methodology used to apply the synthetically tested inversion to real data, using the Thwaites Glacier catchment as our example.

Our base data for surface elevation and velocity were, respectively, the REMA digital elevation model with 8 m resolution (Howat et al., 2019) and output from the NASA MEaSURES ITS-LIVE project with 120 m resolution (Gardner et al., 2018). Based on the latter, we therefore inverted for bed topography and slipperiness at 120 m resolution. An estimate of the ice

thickness in each 50 km by 50 km region is obtained from a 50 km averaged version of Bedmachine Antarctica ice thickness (Morlighem et al., 2020), and this is the only prior information about ice thickness used in the inversion.

In the synthetic tests discussed above, the inversion was run over a single 50 km by 50 km grid, with the outer 10 % of the grid discarded to reduce edge effects introduced during the Fourier transform. To look at the whole Thwaites catchment we

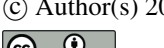


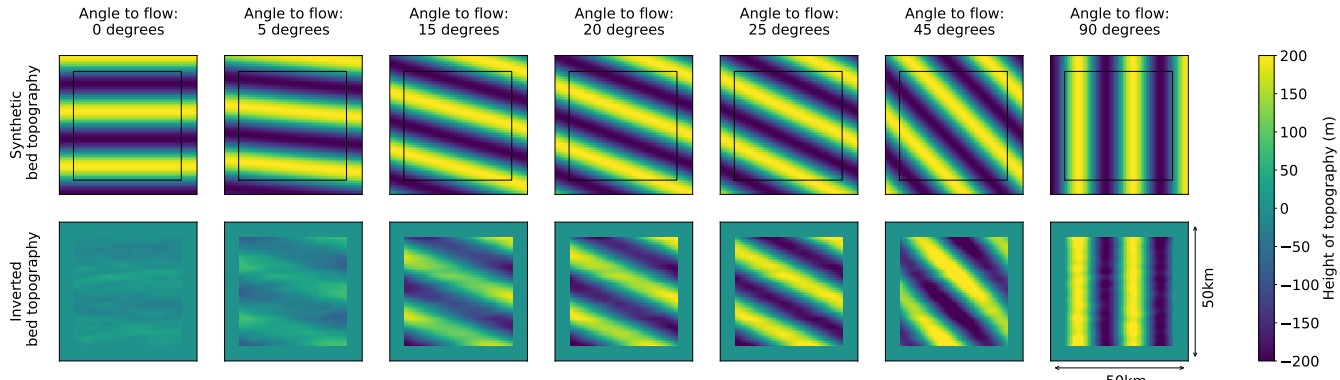

**Figure 1.** The effect of orientation to flow direction on how well landforms (top row; created synthetically) can be resolved by the inversion. These tests are presented on a 50 km by 50 km grid, where in the inversion results (bottom row) the outer 5 km is greyed out to hide edge effects that will be neglected. In these simulations mean ice thickness $\bar{h}$ = 2000 m, mean slipperiness $\bar{C}$ = 100, surface slope $\alpha$ = 0.02, amplitude = 200 m, and wavelength $\lambda$ = 20 km. Landforms at an angle of less than $15°$ to the ice flow direction are not well resolved because they fall in the null space of the inversion.

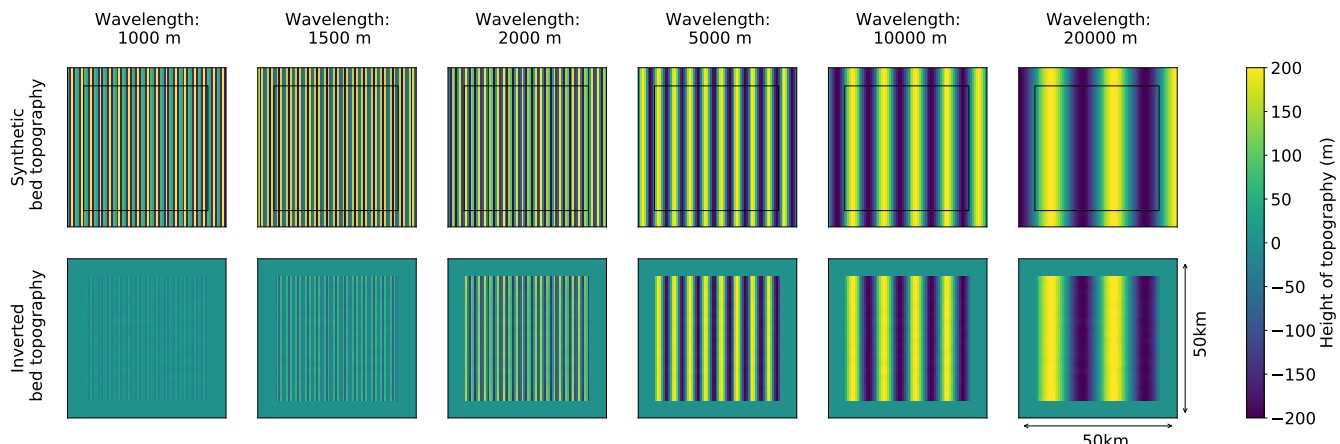

**Figure 2.** The effect of wavelength on how well landforms (top row; created synthetically) can be resolved by the inversion. These tests are presented on a 50 km by 50 km grid, where in the inversion results (bottom row) the outer 5 km is greyed out to hide edge effects that will be neglected. In these simulations mean ice thickness $\bar{h}$ = 2000 m, mean slipperiness $\bar{C}$ = 100, surface slope $\alpha$ = 0.02, amplitude = 200 m, and angle to flow $\theta$ = $90°$. Landforms with a wavelength of less than 2 km can not be well resolved, due to the shallow ice-stream approximation.



**Figure 3.** The effect of amplitude on how well landforms and slipperiness (rows 1 and 3, respectively; created synthetically) can be resolved by the inversion. These tests are presented on a 50 km by 50 km grid, where in the inversion results (rows 2 and 4, respectively) the outer 5 km is greyed out to hide edge effects will should be neglected. In these simulations mean ice thickness $\bar{h}$ = 2000 m, mean slipperiness $\bar{C}$ = 100, surface slope $\alpha$ = 0.02, wavelength $\lambda$ = 20 km and angle to flow $\theta$= 60°. The limiting factor on how well small amplitudes can be resolved is errors in the ice-surface data, which are simulated by adding noise to the synthetic surface generated from the synthetic bed. Noise is added with an amplitude of 2 m for the ice surface elevation (Howat et al., 2019), and an amplitude of 15 ms$^{-1}$ for the ice surface velocity (Gardner et al., 2018). Bedforms with an amplitude of less than 10 m, and slipperiness of less than around $1 \times 10^{-4}$ m yr$^{-1}$ Pa$^{-1}$ are not well resolved.



could use a set of adjacent 50 km by 50 km grids, but instead we chose to use more densely distributed grids which overlap.
This is because in the region of overlap, the variability in the inverted results is a measure of how well the physics that we
have assumed fits reality. Since the dominant approximation made in the physics is linearisation, we assume that variability in
regions of overlap is mainly a measure of non-linearity in the physics of ice flow, although there could also be a contribution
from the inappropriate use of the shallow-ice approximation. Variability is calculated as the standard deviation of overlapping
solutions at each grid point. Discarding more of the outer part of each overlapping region further reduces edge effects. Figure
4a summarises this methodology. The bed topography and slipperiness results presented here are the grid-point by grid-point
means of nine overlapping grids where each overlapping region is 1.67 km by 1.67 km. These values were chosen following
tests on a small region of the real data (Figure 4b).

When applied to real surface-elevation and velocity data, this method generates four products: the mean and standard deviation of bed topography and the mean and standard deviation of bed slipperiness.

## 3 Results for Thwaites Glacier

Figure 5 shows the bed conditions we inverted from REMA (Howat et al., 2019) and ITSLIVE (Gardner et al., 2018) over a
280 km by 160 km region of the main trunk of Thwaites Glacier.

The bed topography product from the inversion is shown in Figure 5b. On the basin scale, the main basal topographic features
identified by the inversion are several sets of parallel ridges which are oriented perpendicular to the direction of ice flow, and
smooth basins in between these ridges. The location of these ridges matches well with the BedMachine Antarctica bed (Figure
5a), particularly around the subglacial lakes, which appear to be between successive sets of ridges. The smoother topography
in the basins between ridges is reflected in the inversion, particularly in the basin to the east of the Upper Thwaites region
(Basin Y, Figure 5). Many smaller hills also match Bedmachine Antarctica, such as those at the upstream (south) end of the
Upper Thwaites radar grid. However, the inversion also generates some notable features that are not present in the Bedmachine
Antarctica bed, such as the north-eastern extent of the central ridge next to the most upstream subglacial lake (Ridge Z, Figure
5b).

The standard deviation of the bed topography (Figure 5c) represents the range in model outputs from overlapping grid squares
which use different regions of the ice surface. As might be expected, this standard deviation is lower in the central trunk of the
glacier where the topographic gradients are smaller. The standard deviation is higher at the edges of the glacier trunk where the
gradient of the topography changes, and the shallow-ice-stream approximation breaks down. Standard deviation is also high in
the north-west part of the inversion where some of the input surface is the surface of the floating ice shelf, which is subject to
different physical processes than the grounded ice. The inversion does not produce results for the north-west corner as there
are gaps in the input data where the surface sampled is not ice, but open ocean.

The bed slipperiness product from the inversion is shown in Figure 5d. The pattern in the slipperiness output is similar to
the topography, with a dominant east to west lineation, although it is slightly difficult to make out due to the strong underlying
slipperiness variation across the Thwaites Glacier region. This directional trend in slipperiness is also observed in the slipperi-



**Figure 4.** a) Multiple overlapping grids are used in the inversion when applied to real data to allow the variability between solutions to be studied. b) The effect of changing the number of overlapping grids and the fraction of the overlapping central region used to calculate the standard deviation of bed topography from the inversion in a roughly 60 km by 60 km region.



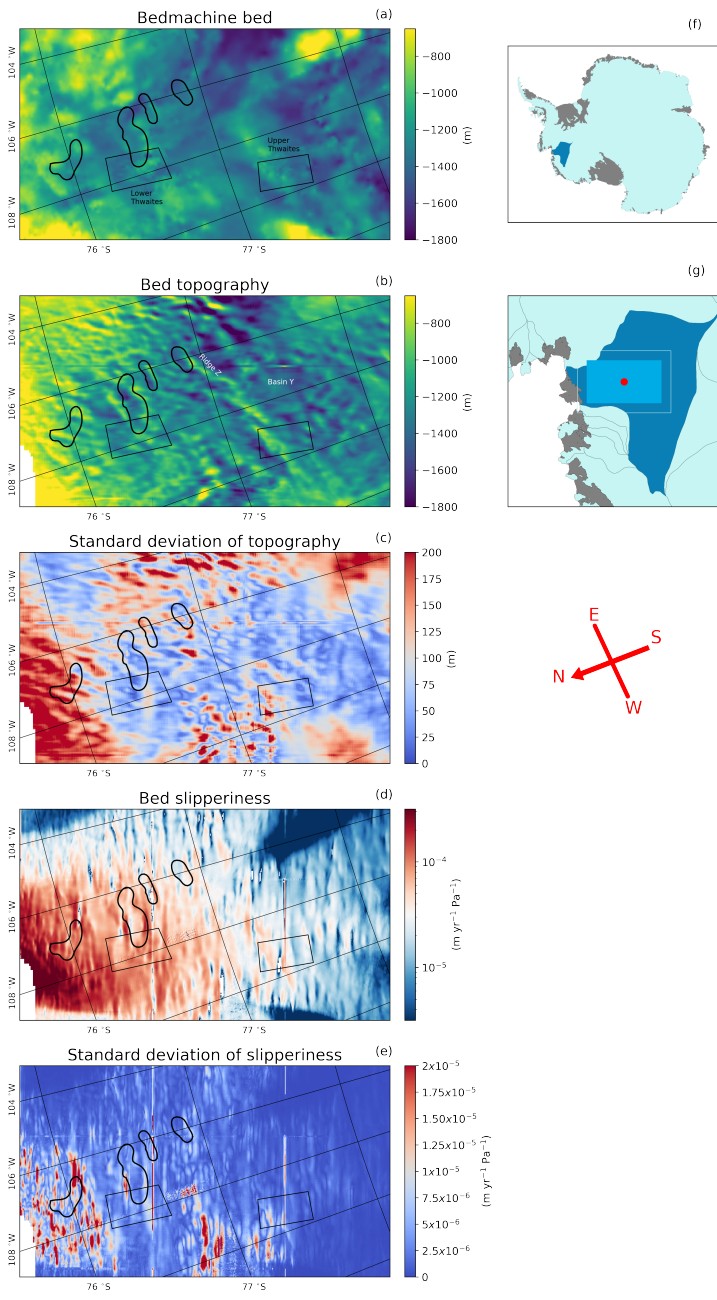

**Figure 5.** Inversion outputs across a 160 km by 280 km region of Thwaites Glacier (location shown in panels f) and g)). The compass directions shown are accurate at the center of the region (red dot in panel g), but may vary by up to 10 degrees across the region due to the polar stereographic projection. a) Bed topography from BedMachine Antarctica (Morlighem et al., 2020), and b) bed topography, c) standard deviation of bed topography, d) bed slipperiness and e) standard deviation of bed slipperiness from our inversions of REMA (Howat et al., 2019) and ITSLIVE (Gardner et al., 2018) at 120 m resolution. Black rectangles depict regions of pre-existing highly resolved bed topography (Holschuh et al., 2020) examined in Figure 6. The four subglacial lakes observed from surface altimetry changes by Smith et al. (2017) are also outlined in black.





ness output of other inversions in the Thwaites Glacier region (Barnes et al., 2021). There is less variability in slipperiness in the basins where the topography is smoother. In contrast the ridges have much more variable slipperiness. Like the standard deviation of bed topography, the standard deviation of bed slipperiness (Figure 5e) is highest around the edges of the central
trunk where there are higher topographic gradients.

## 3.1   Comparison to radar data

In order to assess the success of the inversion, the bed topography output can be compared to existing radar data. Figure 6 shows a comparison between the inverted bed topography and bed topography sounded by swath radar at sub-ice thickness resolution across two 20 km by 40 km regions (Holschuh et al., 2020). The inverted bed shows a good match to the swath-
radar-imaged bed at larger scales, picking out the locations of all the main hills and valleys. There is a better match for the Upper Thwaites region than the Lower Thwaites region, and the fact that the inversion detects the channel between the ridges in the downstream (left) part of Upper Thwaites is particularly encouraging.

    We further compare the inverted bed topography with the bed topography sounded by swath-radar along ice flow (Figure 7) and across ice flow (Figure 8) by airborne radar over Thwaites Glacier in the 2019/2020 field season (Jordan and Robinson,
2021). Further comparisons to more radar flight lines collected in the same field season can be seen in the supplementary information. These figures also demonstrate that the inversion performs well in detecting the main hills and valleys, but also highlights that their amplitudes are not always resolved correctly. This is likely due to variability in the local mean slipperiness away from the imposed global value of non-dimensional slipperiness $\bar{C} = 100$. If a non-dimensional slipperiness $\bar{C} = 150$ is imposed (Figures 6, 7c, 8c) then the amplitudes of the inverted topography are reduced. Sometimes there is also an offset
between the inverted and radar-sounded beds, caused by using a 50 km averaged version of the Bedmachine ice thickness as the inversion ice thickness, rather than more detailed prior information.

**Figure 6.** a) Bed topography acquired at sub-ice-thickness resolution by swath radar (Holschuh et al., 2020); b) our inverted bed topography with mean non-dimensional slipperiness $\bar{C} = 100$; and c) our inverted bed topography with mean slipperiness $\bar{C} = 150$, for site labelled 'Upper Thwaites' in Figure 5a. d), e) and f) show equivalent products for site labelled 'Lower Thwaites' in Figure 5a. The results of a simple linear regression between the swath radar bed and the inverted bed are also given, with $r$ being the regression coefficient and slope the gradient of the line of best fit. A slope of 1 means the amplitude of the inverted bed matches the amplitude of the swath radar bed.



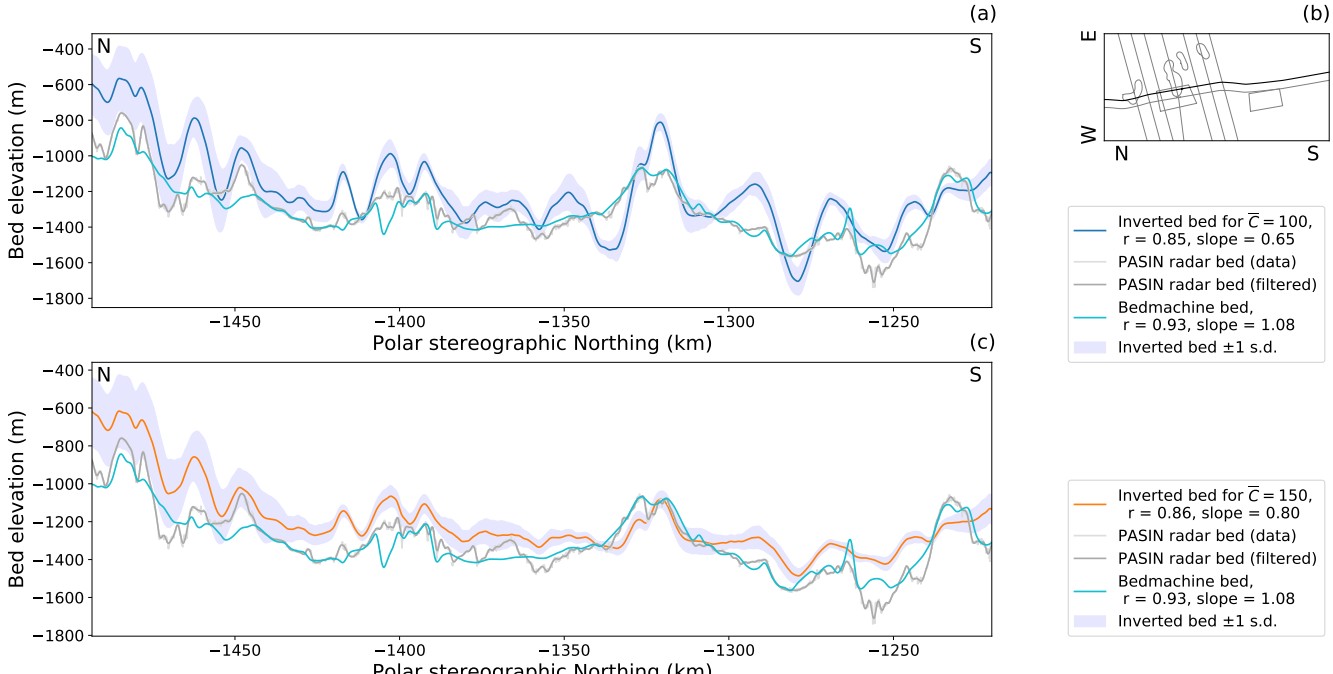

**Figure 7.** a) Comparative plot of inverted bed topography with mean non-dimensional slipperiness $\bar{C} = 100$ with along-flow radar-sounded bed topography. Bed topography is given in three forms: as unfiltered bed picks from the 2019/20 airborne surveys (Jordan and Robinson, 2021); a version of the same filtered to 2 km wavelengths to be more representative of the detail we might expect to image in our inversion; and the bed profile extracted from BedMachine Antarctica (Morlighem et al., 2020). The envelope around the inverted bed topography shows plus or minus one standard deviation. The correlation coefficients ($r$) and slopes given are the results of a linear regression between the inverted bed or the BedMachine Antarctica bed and the filtered radar bed. b) Profile location within the inverted grid. c) As for panel a) but with mean non-dimensional slipperiness $\bar{C} = 150$.



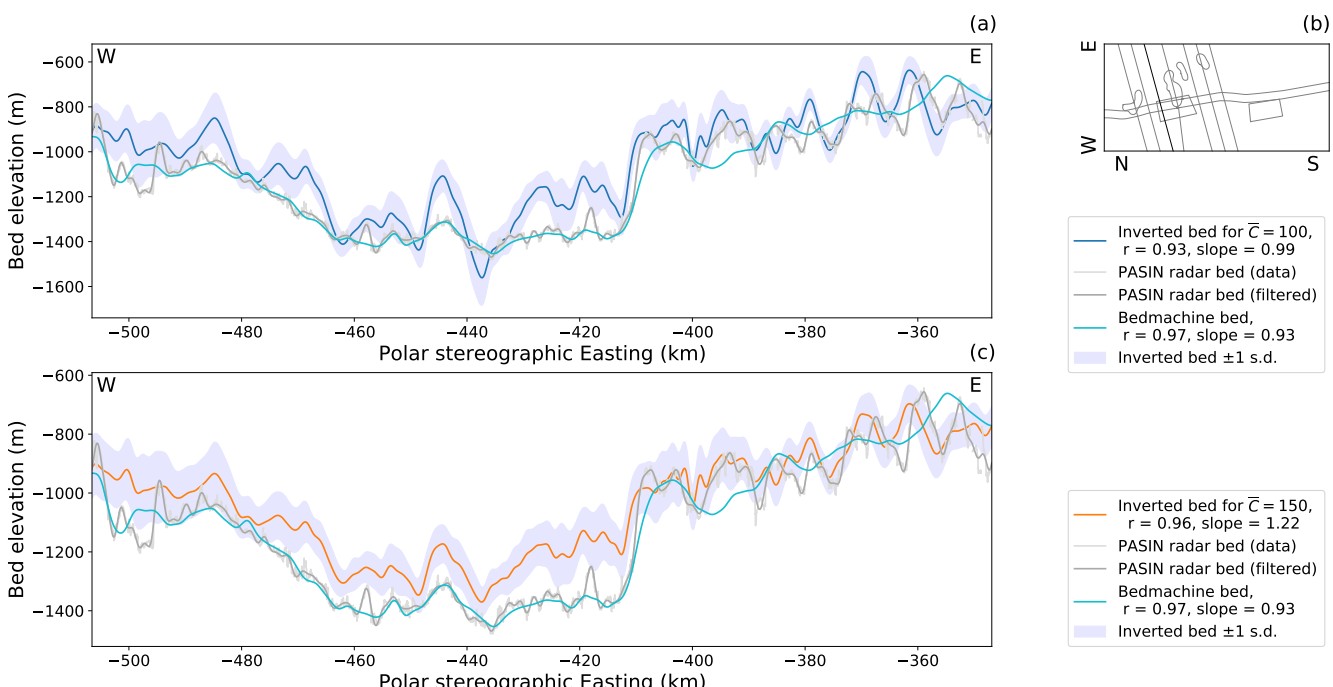

**Figure 8.** a) Comparitive plot of inverted bed topography with mean non-dimensional slipperiness $\bar{C} = 100$ with across flow radar-sounded bed topography. Key as for Figure 7.



## 4   Discussion

Our results demonstrate significant promise for being able to invert for bed topography across parts of Antarctica and other polar regions from surface elevation and velocity data sets. Comparisons with existing radar data available from Thwaites
Glacier suggest that within the central trunk of the glacier the bed features identified by the inversion are normally in the correct locations, but are not always centred around the correct depth. These average depth differences are primarily due to the mean ice thickness used in the inversion, which is a 50 km averaged version of the Bedmachine Antarctica ice thickness (Morlighem et al., 2020). For regions where there are radar flight lines and grids, these radar observations could be used instead of the averaged ice thickness to ensure that the bed depth is correct. For regions where there are very few existing radar
data, this method has the potential in the future to identify obstacles to flow which are significant enough to affect surface ice dynamics, even if there is uncertainty about the local or regional ice thickness.

The regions where the inverted bed deviates significantly from the topography picked from radar surveys are of particular interest in assessing the potential of our inversion. Differences between inverted topography and radar lines are likely to be due to physical processes which are not encapsulated by the shallow ice-stream approximation. One place in which the shallow
ice-stream approximation is known to break down is where the mean slope of the bedrock becomes too steep (Gudmundsson, 2003; Le Meur et al., 2004). The effect of this can be observed around the edges of the central trunk of the glacier, where the topographic slope is steep, and the match between the inverted bed and the radar lines is poor (Figure 5). Gudmundsson (2003) derived a full-system non-hydrostatic momentum balance version of the transfer function used in this work. The full-system approach does not rely on the shallow-ice-stream approximation, and should therefore perform better. Future work using this
method is likely to incorporate these more complex equations.

A further consideration in comparing the inverted bed to real data is the steady-state assumption made when deriving the transfer functions. Without repeat radar measurements for Thwaites Glacier we can not be sure of the stability of the bed. If the bed beneath Thwaites Glacier is changing rapidly, as observed at Rutford Ice Stream (Smith et al., 2007) then the surface may not represent the current bed, but some long term average. However, observations at Pine Island Glacier (Brisbourne et al.,
2017; Davies et al., 2018) suggest that the bed is not changing rapidly there, and it is possible that neighbouring Thwaites Glacier might be behaving similarly. Additionally, the erosion observed at Rutford Ice Stream does not significantly change the shape of the topography on the wavelengths resolved by this inversion. If drumlins or mega-scale glaciation lineations (MSGL) were forming, we would not be able to detect them with this method, as landforms aligned to flow fall in the null space of the inversion. The steady state assumption does not only apply to the bed but also to the ice surface, which becomes
more unstable closer to the grounding line. However, since results in the region immediately adjacent to the grounding line are compromised by the different physics of the ice shelf anyway, this is not a significant concern. We therefore consider the steady-state assumption to be suitable for the purposes of this inversion.

As in any modelling study, it is important to explore the behaviour of the inversion when the parameters chosen are varied. In this inversion there are just four fixed parameters which are not derived from the input data: the sliding law exponent, $m$,
the filtering parameter, $p_{filt}$, the weighting factor $\Sigma_s$ and the mean slipperiness, $\bar{C}$. The filtering parameter $p_{filt}$ (Equation



C4) controls which frequencies are suppressed in the inversion to avoid introducing singularities. Higher values of $p_{filt}$ (closer to 0), will filter out lower frequencies (higher wavelengths), and so a value of $p_{filt} = -2$ is chosen to filter out noisy short wavelengths, while maintaining realistic bed features. The weighting factor $\Sigma_s$ (Equation C2) controls the balance in the inversion between the surface elevation and surface velocity data, with smaller values of $\Sigma_s$ weighting the inversion towards

the surface data. Varying $p_{filt}$ and $\Sigma_s$ for the inversion of the real surface data confirms the choice of values from the synthetic tests ($p_{filt} = -2$, $\Sigma_s = 0.001$) as sensible values which return the best match with real bed data.

There is less certainty over what is the most suitable value for the non-dimensional mean slipperiness $\bar{C}$. Although we have some measurements of the bed properties of Thwaites glacier from seismic lines, gravity and magnetic inversions (Diehl, 2008; Jordan et al., 2010; Muto et al., 2019a), these are spatially limited and it is not currently clear how these properties combine

into slipperiness at the bed (Kyrke-Smith et al., 2017). If $\bar{C}$ is higher, then the amplitude of bed variability in the inversion output falls. Given the geological variability likely to be associated with multiple rifted tectonic blocks (Dunham et al., 2020) and the sediments deposited in those rift basins (Muto et al., 2019a, b) it is unlikely that the mean slipperiness, $\bar{C}$, is the same across the whole region modelled here. Modelling studies which compare the results of different inversion procedures show that slipperiness may be quite variable across the Thwaites Glacier catchment (Barnes et al., 2021). In additional we note that

the trend is quite different from features observed in the inverted topography, showing that the slipperiness map is not a result of linear trade-offs with topography in the inversion solution. The three dimensional radar grids (Holschuh et al., 2020) are both located within regions with more topographic variability, likely unlifted rift blocks. This may explain why a lower value of $\bar{C} = 100$ gives the best match in these regions, whereas a higher value of $\bar{C} = 150$ (more slippery) gives a better match for the radar lines which cover both lithologies. It may also be that the 3D grids, which contain both along and across flow variability,

are more representative of the bed than the 2D radar lines. For this reason, the catchment scale bed topography presented in Figure 5 is from the inversion with $\bar{C} = 100$.

This uncertainty in $\bar{C}$ means that some prior radar information is useful in order to calibrate the inversion method to give the best results. However, changing $\bar{C}$ only alters the amplitude, so even if there is no prior information the inversion will still identify hills and troughs. More detailed analysis of seismic and gravity data alongside the results of the inversion could also

reveal trends that could be useful when applying this technique elsewhere.

The high-resolution swath radar grids (Holschuh et al., 2020) presented here have already been included in Bedmachine Antarctica. Using the radar grids which are currently available, we can not therefore explore how well this inversion method performs compared to Bedmachine Antarctica (Morlighem et al., 2020) over a dense radar grid. Both techniques use ice surface elevation and velocity datasets. Bedmachine Antarctica uses these datasets and the principles of mass conservation (or

streamline diffusion in slow moving areas) to interpolate between detailed prior information on ice thickness from existing radar measurements. In contrast, the inversion method only requires an estimate of average ice thickness for each 50 km by 50 km grid and uses the linear perturbation theory described. This 50 km averaged ice thickness is subtracted from the ice surface to provide a reference bed to which the inversion adds perturbations, as shown in Figure 9. However, even if a single ice-thickness ($\bar{h}$) value is used for the entire catchment, then the inversion method will still identify the location of hills and

troughs in the bed topography, although the amplitudes and absolute depths of these features may be affected. Since the mass





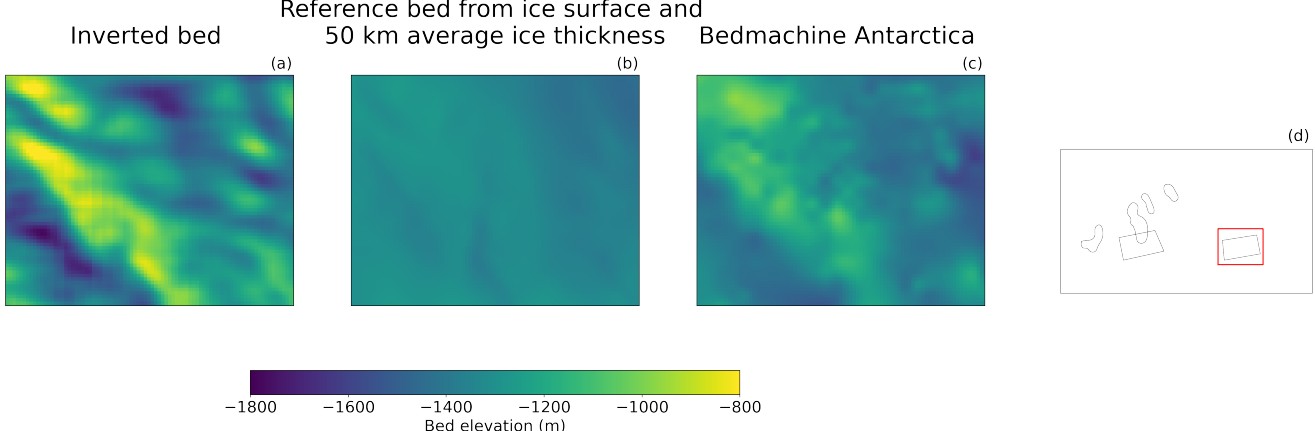

**Figure 9.** a) The inverted bed topography, b) The reference bed topography (calculated from the ice surface with the 50-km-averaged ice surface subtracted) to which the inversion adds perturbations, and c) the Bedmachine Antarctica bed topography (Morlighem et al., 2020) for the region containing the Upper Thwaites radar grid (Holschuh et al., 2020). Panel d) shows in red the outline of this region within the inverted grid shown in Figure 5, and the locations of four subglacial lakes (Smith et al., 2017) and the two radar grids (Holschuh et al., 2020)
.

conservation method used in Bedmachine assesses the ice flow through a series of flux gates ideally constrained by topography, it is much more reliant on good-quality, closely-spaced, ice thickness measurements from radar systems.

A comparison of the two methods over an area where radar data have not yet been incorporated into Bedmachine would allow an assessment of the reliability of the two techniques, and identification of any artificial bed features introduced by each.
Since the two radar grids presented (Holschuh et al., 2020) were included in the derivation of BedMachine Antarctica, no independent test is possible until more radar grids are collected.

## 5   Conclusions

We present the method and results of an inversion of ice surface elevation and velocity for bed topography and slipperiness in the Thwaites Glacier region. Our method builds on the method used by Thorsteinsson et al. (2003) in their study of MacAyeal
Ice Stream, but is based on a steady-state linear perturbation analysis of the shallow-ice-stream equations (MacAyeal, 1989; Gudmundsson, 2003). Synthetic tests show that this method can resolve variability in bed topography and slipperiness on wavelengths greater than one ice thickness, and at amplitudes of more than 10 m for topography or $1 \, \mathrm{x}10^{-4} \, \mathrm{myr}^{-1}\mathrm{Pa}^{-1}$ for slipperiness, as long as the variability is not aligned to the ice flow direction. Comparison of the results of the inversion with radar grids and flight lines suggests that the inversion does a good job of picking up features in the bed, with the correlation
coefficient of a linear regression between the inverted bed and the radar bed as high as $r = 0.93$ along some flight lines. This method works best in the central trunk of the glacier, where the gradient of the long wavelength topography is low and relatively constant. Mismatches between the inverted bed topography and radar measurements are probably due to one of three factors:





an incorrect ice thickness for that region, an unusually sticky or slippery bed, or physical processes not accounted for by the steady-state linearised shallow-ice stream approximation. Future work, including incorporating more local prior ice thickness

data from radar measurements, and the non-hydrostatic transfer functions from Gudmundsson (2003), may help to reduce these mismatches. Overall, the inversion provides an additional tool for studying landforms in the bed beneath glaciers which have a significant impact on ice flow. It will be particularly useful in ice streams where radar flight lines are sparse and standard interpolation techniques struggle, potentially reducing uncertainties in modelling the future behaviour of those regions and their contributions to global sea level rise.

*Code and data availability.* The output data from the inversion is available on Zenodo at https://doi.org/10.5281/zenodo.5105687. The code for the inversion and plotting the figures is available on Zenodo at https://doi.org/10.5281/zenodo.5494600. The surface elevation data from REMA (Howat et al., 2019) and velocity data from ITS_LIVE (Gardner et al., 2018) used as inputs in the inversion are available freely online, as are the swath radar (Holschuh et al., 2020), airborne radar (Jordan and Robinson, 2021) and Bedmachine Antarctica (Morlighem et al., 2020) bed datasets to which the results of the inversion are compared.

## 350 Appendix A: Derivation of transfer functions from a topography perturbation

Following Gudmundsson (2008), we start with the shallow-ice-stream equations (MacAyeal, 1989),

$$\partial_x(4h\eta\partial_x u + 2h\eta\partial_y v) + \partial_y(h\eta(\partial_x v + \partial_y u)) - (u/c)^{1/m} = \rho gh\partial_x(s)\cos\alpha - \rho gh\sin\alpha \tag{A1}$$

$$\partial_y(4h\eta\partial_y v + 2h\eta\partial_x u) + \partial_x(h\eta(\partial_y u + \partial_x v)) - (v/c)^{1/m} = \rho gh\partial_y(s)\cos\alpha \tag{A2}$$

where $u$, $v$, $w$ are the velocity components in the $x, y$ and $z$ directions respectively, $h$ is the ice thickness, $\eta$ is the effective

ice viscosity, $c$ is the basal slipperiness, $m$ is a sliding law parameter, $\rho$ is the ice density, $g$ is the acceleration due to gravity, $s$ is the ice surface elevation, $b$ is the ice bed elevation, and $\alpha$ is the mean ice surface slope in the $x$ direction.

We consider the response to a small perturbation in basal slipperiness, $b$, linearising with $h = \bar{h} + \Delta h$, $s = \bar{s} + \Delta s$, $b = \bar{b} + \Delta b$, $u = \bar{u} + \Delta u$, $v = \Delta v$, $w = \Delta w$ and $c = \bar{c}$.

This gives the first order equations

$$4\eta\bar{h}\partial_{xx}^2\Delta u + 3\eta\bar{h}\partial_{xy}^2\Delta v + \eta\bar{h}\partial_{yy}^2\Delta u - \gamma\Delta u = \rho g\bar{h}\cos\alpha\partial_x\Delta s - \rho g\sin\alpha\Delta h \tag{A3}$$

$$4\eta\bar{h}\partial_{yy}^2\Delta v + 3\eta\bar{h}\partial_{xy}^2\Delta u + \eta\bar{h}\partial_{xx}^2\Delta v - \gamma\Delta v = \rho g\bar{h}\cos\alpha\partial_y\Delta s \tag{A4}$$

To the first order, and importantly in the steady state, we also have the upper and lower boundary conditions

$$\bar{u}\partial_x\Delta s - \Delta w(s) = 0 \tag{A5}$$

$$\bar{u}\partial_x\Delta b - \Delta w(b) = 0 \tag{A6}$$





All variables are then Fourier transformed with respect to the spatial variables $x$ and $y$. In the forward Fourier transform the two space variables both carry a positive sign and the wavenumbers in the $x$ and $y$ directions are denoted by $k$ and $l$ respectively. This Fourier transform gives

$$4\eta\bar{h}k^2\hat{u} + 3\eta\bar{h}kl\hat{v} + \eta\bar{h}l^2\hat{u} + \gamma\hat{u} = \rho g\bar{h}\cos\alpha ik\hat{s} + \rho g\sin\alpha\hat{h} \tag{A7}$$

$$4\eta\bar{h}l^2\hat{v} + 3\eta\bar{h}kl\hat{u} + \eta\bar{h}k^2\hat{v} + \gamma\hat{v} = \rho g\bar{h}\cos\alpha il\hat{s} \tag{A8}$$

$$\hat{w}(\bar{s}) = -i\bar{u}k\hat{s} \tag{A9}$$

$$\hat{w}(\bar{b}) = -i\bar{u}k\hat{b} \tag{A10}$$

where $\hat{h} = \hat{s} - \hat{b}$.

From depth integration of the Fourier-transformed incompressibility condition $w_z + u_x + v_y = 0$ we have

$$i\bar{h}(k\hat{u} + l\hat{v}) = \hat{w}(\bar{s}) - \hat{w}(\bar{b}) \tag{A11}$$

which, along with the steady-state boundary conditions, yields

$$i\bar{h}(k\hat{u} + l\hat{v}) = -ik\bar{u}\hat{s} + ik\bar{u}\hat{b}. \tag{A12}$$

Equations A7, A8 and A12 form a linear system of equations in $\hat{s}$, $\hat{u}$, $\hat{v}$ and $\hat{b}$ which can be solved algebraically using standard techniques:

$$\begin{bmatrix} (3\eta\bar{h}k^2 + \nu) & (3\eta\bar{h}kl) \\ (3\eta\bar{h}kl) & (3\eta\bar{h}l^2 + \nu) \end{bmatrix} \begin{bmatrix} \hat{u} \\ \hat{v} \end{bmatrix} = \begin{bmatrix} \left(ik\tau_d\cot\alpha + \frac{\tau_d}{\bar{h}}\right)\hat{s} - \frac{\tau_d}{\bar{h}}\hat{b} \\ il\tau_d\cot\alpha\hat{s} \end{bmatrix}$$

The determinant of the left-hand side of these equations:

$$\left(3\eta\bar{h}k^2 + \nu\right)\left(3\eta\bar{h}l^2 + \nu\right) - \left(3\eta\bar{h}kl\right)\left(3\eta\bar{h}kl\right)$$
$$= 9\eta^2\bar{h}^2k^2l^2 + 3\eta\bar{h}l^2\nu + 3\eta\bar{h}k^2\nu + \nu^2 - 9\eta^2\bar{h}^2k^2l^2$$
$$= 3\eta\bar{h}j^2\nu + \left(\bar{h}\eta j^2 + \gamma\right)\nu$$
$$= \left(4\bar{h}\eta j^2 + \gamma\right)\nu$$
$$= \xi\nu$$

where the following abbreviations have been used to make the algebra clearer to follow: $\nu = \bar{h}\eta j^2 + \gamma$, $j^2 = l^2 + k^2$, and $\xi = 4\bar{h}\eta j^2 + \gamma$.

$$\begin{bmatrix} \hat{u} \\ \hat{v} \end{bmatrix} = \frac{1}{\xi\nu}\begin{bmatrix} (3\eta\bar{h}l^2 + \nu) & (-3\eta\bar{h}kl) \\ (-3\eta\bar{h}kl) & (3\eta\bar{h}k^2 + \nu) \end{bmatrix} \begin{bmatrix} \left(ik\tau_d\cot\alpha + \frac{\tau_d}{\bar{h}}\right)\hat{s} - \frac{\tau_d}{\bar{h}}\hat{b} \\ il\tau_d\cot\alpha\hat{s} \end{bmatrix}$$





Therefore we have:

$$\hat{u} = \frac{1}{\xi\nu}\left((3\eta\bar{h}l^2+\nu)\left(\left(ik\tau_d\cot\alpha+\frac{\tau_d}{\bar{h}}\right)\hat{s}-\frac{\tau_d}{\bar{h}}\hat{b}\right)+(-3\eta\bar{h}kl)(il\tau_d\cot\alpha\hat{s})\right)$$

$$\hat{v} = \frac{1}{\xi\nu}\left((-3\eta\bar{h}kl)\left(\left(ik\tau_d\cot\alpha+\frac{\tau_d}{\bar{h}}\right)\hat{s}-\frac{\tau_d}{\bar{h}}\hat{b}\right)+(3\eta\bar{h}k^2+\nu)(il\tau_d\cot\alpha\hat{s})\right)$$

which simplifies to:

$$\hat{u} = \frac{1}{\xi\nu\bar{h}}\left(3\eta\bar{h}l^2\tau_d\hat{s}-3\eta\bar{h}l^2\tau_d\hat{b}+ik\nu\bar{h}\tau_d\cot\alpha\hat{s}+\tau_d\nu\hat{s}-\tau_d\nu\hat{b}\right) \tag{A13}$$

$$\hat{v} = \frac{1}{\xi\nu\bar{h}}\left(-3\eta\bar{h}kl\tau_d\hat{s}+3\eta\bar{h}kl\tau_d\hat{b}+i\bar{h}l\nu\tau_d\cot\alpha\hat{s}\right) \tag{A14}$$

We then have:

$$i\bar{h}(k\hat{u}+l\hat{v}) = \frac{i}{\xi\nu}\left(k(3\eta\bar{h}l^2\tau_d\hat{s}-3\eta\bar{h}l^2\tau_d\hat{b}+ik\nu\bar{h}\tau_d\cot\alpha\hat{s}+\tau_d\nu\hat{s}-\tau_d\nu\hat{b})\right.$$

$$\left.+l(-3\eta\bar{h}kl\tau_d\hat{s}+3\eta\bar{h}kl\tau_d\hat{b}+i\bar{h}l\nu\tau_d\cot\alpha\hat{s})\right)$$

$$= \frac{i}{\xi\nu}\left(ik^2\nu\bar{h}\tau_d\cot\alpha\hat{s}+k\tau_d\nu\hat{s}-k\tau_d\nu\hat{b}+i\bar{h}l^2\nu\tau_d\cot\alpha\hat{s}\right)$$

$$= \frac{1}{\xi}\left(-j^2\bar{h}\tau_d\cot\alpha\hat{s}+ik\tau_d\hat{s}-ik\tau_d\hat{b}\right)$$

In the steady state, the kinematic boundary condition is

$$i\bar{h}(k\hat{u}+l\hat{v}) = \hat{w}(s)-\hat{w}(b)$$

$$= -i\bar{u}k\hat{s}+i\bar{u}k\hat{b}$$

Substituting the expression from above:

$$\frac{1}{\xi}\left(-j^2\bar{h}\tau_d\cot\alpha\hat{s}+ik\tau_d\hat{s}-ik\tau_d\hat{b}\right) = -i\bar{u}k\hat{s}+i\bar{u}k\hat{b}$$

Rearranging

$$\left(i\bar{u}k+\frac{ik\tau_d}{\xi}-\frac{j^2\bar{h}\tau_d\cot\alpha}{\xi}\right)\hat{s} = \frac{1}{\xi}\left(ik\tau_d+ik\bar{u}\xi\right)\hat{b}$$

$$\left(ik(\bar{u}+\frac{\tau_d}{\xi})-\frac{j^2\bar{h}\tau_d\cot\alpha}{\xi}\right)\hat{s} = \frac{1}{\xi}\left(ik(\tau_d+\bar{u}\xi)\right)\hat{b}$$

$$\xi p\hat{s} = ik(\tau_d+\bar{u}\xi)\hat{b}$$





In agreement with Gudmundsson (2008), this leads to the steady-state transfer function

$$T_{sb}(k,l,t) = \frac{\hat{s}}{\hat{b}} = \frac{ik(\bar{u}\xi + \tau_d)}{p\xi} \qquad (A15)$$

Expanding the expression for $\hat{u}$:

$$\hat{u} = \frac{1}{\xi\nu\bar{h}}\left(3\eta\bar{h}l^2\tau_d\hat{s} - 3\eta\bar{h}l^2\tau_d\hat{b} + ik\nu\bar{h}\tau_d\cot\alpha\hat{s} + \tau_d\nu\hat{s} - \tau_d\nu\hat{b}\right) \qquad (A13 \text{ revisited})$$

$$= \frac{\tau_d}{\xi\nu\bar{h}}\left(\bar{h}\nu(ik\cot\alpha)\hat{s} + (3\eta\bar{h}l^2 + \nu)\hat{s} - (3\eta\bar{h}l^2 + \nu)\hat{b}\right)$$

$$= \frac{\tau_d}{\xi\nu\bar{h}}\left(\bar{h}\nu(ik\cot\alpha)\left(\frac{ik(\bar{u}\xi + \tau_d)}{p\xi}\right)\hat{b} + (3\eta\bar{h}l^2 + \nu)\left(\frac{ik(\bar{u}\xi + \tau_d)}{p\xi}\right)\hat{b} - (3\eta\bar{h}l^2 + \nu)\hat{b}\right)$$

$$= \frac{\tau_d}{\xi\nu\bar{h}p\xi}\left(\bar{h}\nu(ik\cot\alpha)\left(ik(\bar{u}\xi + \tau_d)\right) + (3\eta\bar{h}l^2 + \nu)\left(ik(\bar{u}\xi + \tau_d)\right) - (3\eta\bar{h}l^2 + \nu)p\xi\right)\hat{b}$$

$$= \frac{\tau_d}{\xi\nu\bar{h}p\xi}\left(\bar{h}\nu(ik\cot\alpha)\left(ik(\bar{u}\xi + \tau_d)\right) + (3\eta\bar{h}l^2 + \nu)\left(ik(\bar{u}\xi + \tau_d)\right)\right.$$

$$\left. - (3\eta\bar{h}l^2 + \nu)\left(ik(\bar{u}\xi + \tau_d) - j^2\tau_d\bar{h}\cot\alpha\right)\right)\hat{b}$$

$$= \frac{\tau_d}{\xi\nu\bar{h}p\xi}\left(\bar{h}\nu(ik\cot\alpha)\left(ik(\bar{u}\xi + \tau_d)\right) - (3\eta\bar{h}l^2 + \nu)\left(-j^2\tau_d\bar{h}\cot\alpha\right)\right)\hat{b}$$

$$= \frac{\tau_d\cot\alpha}{\xi\nu p\xi}\left(-k^2\nu\bar{u}\xi - \nu k^2\tau_d + 3\eta\bar{h}j^2l^2\tau_d + \nu j^2\tau_d\right)\hat{b}$$

$$= \frac{\tau_d\cot\alpha}{\xi\nu p\xi}\left(3\eta\bar{h}j^2l^2\tau_d + \nu l^2\tau_d - k^2\bar{u}\xi\right)\hat{b}$$

$$= \frac{\tau_d\cot\alpha}{\xi\nu p\xi}\left(\xi l^2\tau_d - k^2\bar{u}\xi\right)\hat{b}$$

$$\hat{u} = \frac{\tau_d\cot\alpha}{\xi\nu p}\left(l^2\tau_d - k^2\bar{u}\right)\hat{b}$$

remembering that $\xi = 4\eta\bar{h}j^2 + \nu$.

In agreement with Gudmundsson (2008), this leads to the steady-state transfer function:

$$T_{ub}(k,l,t) = \frac{\hat{u}}{\hat{b}} = \frac{\tau_d\cot\alpha(l^2\tau_d - k^2\bar{u})}{\xi\nu p} \qquad (A16)$$





Expanding the expression for $\hat{v}$:

$$\hat{v} = \frac{1}{\xi\nu\bar{h}}\left(-3\eta\bar{h}kl\tau_d\hat{s} + 3\eta\bar{h}kl\tau_d\hat{b} + i\bar{h}l\nu\tau_d\cot\alpha\hat{s}\right)$$

$$= \frac{\tau_d l}{\xi\nu}\left(-3\eta k\left(\frac{ik(\bar{u}\xi+\tau_d)}{p\xi}\right)\hat{b} + 3\eta k\hat{b} + i\nu\cot\alpha\left(\frac{ik(\bar{u}\xi+\tau_d)}{p\xi}\right)\hat{b}\right)$$

$$= \frac{\tau_d l}{\xi\nu p\xi}\left(-3\eta k\left(ik(\bar{u}\xi+\tau_d)\right) + 3\eta kp\xi + i\nu\cot\alpha\left(ik(\bar{u}\xi+\tau_d)\right)\right)\hat{b}$$

$$= \frac{\tau_d l}{\xi\nu p\xi}\left(-3\eta k\left(ik(\bar{u}\xi+\tau_d)\right) + 3\eta k\left(ik(\bar{u}\xi+\tau_d) - j^2\tau_d\bar{h}\cot\alpha\right) + i\nu\cot\alpha\left(ik(\bar{u}\xi+\tau_d)\right)\right)\hat{b}$$

$$= \frac{\tau_d lk}{\xi\nu p\xi}\left(+3\eta\left(-j^2\tau_d\bar{h}\cot\alpha\right) - \nu\cot\alpha(\bar{u}\xi+\tau_d)\right)\hat{b}$$

$$= -\frac{\tau_d lk\cot\alpha}{\xi\nu p\xi}\left(-(3\eta\bar{h}j^2+\nu)\tau_d - \nu\bar{u}\xi\right)\hat{b}$$

$$\hat{v} = \frac{\tau_d lk\cot\alpha}{\xi\nu p}\left(\tau_d + \nu\bar{u}\right)\hat{b}$$

remembering that $\xi = 4\eta\bar{h}j^2 + \nu$.

In agreement with Gudmundsson (2008), this leads to the steady-state transfer function:

$$T_{vb}(k,l,t) = \frac{\hat{v}}{\hat{b}} = \frac{kl\tau_d\cot\alpha(\tau_d+\nu\bar{u})}{\xi\nu p} \tag{A17}$$

## Appendix B: Derivation of transfer functions from a slipperiness perturbation

Starting once again with the shallow-ice-stream equations (Equations A1 and A2; MacAyeal, 1989), this time we consider the response to a small perturbation in basal slipperiness, $c$, linearising with $h = \bar{h} + \Delta s$, $s = \bar{s} + \Delta s$, $b = \bar{b}$, $u = \bar{u} + \Delta u$, $v = \Delta v$, $w = \Delta w$ and $c = \bar{c}(1 + \Delta c)$ where $\Delta c$ is the fractional slipperiness.

This gives the first order equations

$$4\eta\bar{h}\partial_{xx}^2\Delta u + 3\eta\bar{h}\partial_{xy}^2\Delta v + \eta\bar{h}\partial_{yy}^2\Delta u - \gamma\Delta u = \rho g\bar{h}\cos\alpha\partial_x\Delta s - \rho g\sin\alpha\Delta s - \gamma\bar{u}\Delta c \tag{B1}$$

$$4\eta\bar{h}\partial_{yy}^2\Delta v + 3\eta\bar{h}\partial_{xy}^2\Delta u + \eta\bar{h}\partial_{xx}^2\Delta v - \gamma\Delta v = \rho g\bar{h}\cos\alpha\partial_y\Delta s \tag{B2}$$

Fourier transforming with respect to the spatial variables $x$ and $y$ gives:

$$4\eta\bar{h}k^2\hat{u} + 3\eta\bar{h}kl\hat{v} + \eta\bar{h}l^2\hat{u} + \gamma\hat{u} = \rho g\bar{h}\cos\alpha ik\hat{s} + \rho g\sin\alpha\hat{s} + \gamma\bar{u}\hat{c} \tag{B3}$$

$$4\eta\bar{h}l^2\hat{v} + 3\eta\bar{h}kl\hat{u} + \eta\bar{h}k^2\hat{v} + \gamma\hat{v} = \rho g\bar{h}\cos\alpha il\hat{s} \tag{B4}$$





As there is no bed topography perturbation, the steady-state boundary conditions become

$$i\bar{h}(k\hat{u} + l\hat{v}) = -ik\bar{u}\hat{s}. \tag{B5}$$

Equations B3, B4 and B5 form a linear system of equations which can be solved using standard algebraic techniques:

$$\begin{bmatrix} (3\eta\bar{h}k^2 + \nu) & (3\eta\bar{h}kl) \\ (3\eta\bar{h}kl) & (3\eta\bar{h}l^2 + \nu) \end{bmatrix} \begin{bmatrix} \hat{u} \\ \hat{v} \end{bmatrix} = \begin{bmatrix} \left(ik\tau_d\cot\alpha + \frac{\tau_d}{h}\right)\hat{s} + \gamma\bar{u}\hat{c} \\ il\tau_d\cot\alpha\hat{s} \end{bmatrix}$$

The determinant of the left-hand side:

$$\left(3\eta\bar{h}k^2 + \nu\right)\left(3\eta\bar{h}l^2 + \nu\right) - \left(3\eta\bar{h}kl\right)\left(3\eta\bar{h}kl\right)$$
$$= 9\eta^2\bar{h}^2k^2l^2 + 3\eta\bar{h}l^2\nu + 3\eta\bar{h}k^2\nu + \nu^2 - 9\eta^2\bar{h}^2k^2l^2$$
$$= 3\eta\bar{h}j^2\nu + \left(\bar{h}\eta j^2 + \gamma\right)\nu$$
$$= \left(4\bar{h}\eta j^2 + \gamma\right)\nu$$
$$= \xi\nu$$

where the following abbreviations have been used to make the algebra easier to follow: $j^2 = l^2 + k^2$, $\xi = 4\bar{h}\eta j^2 + \gamma$ and $\nu = \bar{h}\eta j^2 + \gamma$.

$$\begin{bmatrix} \hat{u} \\ \hat{v} \end{bmatrix} = \frac{1}{\xi\nu} \begin{bmatrix} (3\eta\bar{h}l^2 + \nu) & (-3\eta\bar{h}kl) \\ (-3\eta\bar{h}kl) & (3\eta\bar{h}k^2 + \nu) \end{bmatrix} \begin{bmatrix} \left(ik\tau_d\cot\alpha + \frac{\tau_d}{h}\right)\hat{s} + \gamma\bar{u}\hat{c} \\ il\tau_d\cot\alpha\hat{s} \end{bmatrix}$$

Therefore we have:

$$\hat{u} = \frac{1}{\xi\nu}\left((3\eta\bar{h}l^2 + \nu)\left(\left(ik\tau_d\cot\alpha + \frac{\tau_d}{h}\right)\hat{s} + \gamma\bar{u}\hat{c}\right) + (-3\eta\bar{h}kl)(il\tau_d\cot\alpha\hat{s})\right)$$

$$\hat{v} = \frac{1}{\xi\nu}\left((-3\eta\bar{h}kl)\left(\left(ik\tau_d\cot\alpha + \frac{\tau_d}{h}\right)\hat{s} + \gamma\bar{u}\hat{c}\right) + (3\eta\bar{h}k^2 + \nu)(il\tau_d\cot\alpha\hat{s})\right)$$

Which simplifies to:

$$\hat{u} = \frac{1}{\xi\nu}\left(\left(3\eta l^2\tau_d + \nu ik\tau_d\cot\alpha + \frac{\nu\tau_d}{\bar{h}}\right)\hat{s} + \left(3\eta\bar{h}l^2 + \nu\right)\gamma\bar{u}\hat{c}\right) \tag{B6}$$

$$\hat{v} = \frac{1}{\xi\nu}\left(-3\eta kl\tau_d\hat{s} + \nu il\tau_d\cot\alpha\hat{s} - 3\eta\bar{h}kl\gamma\bar{u}\hat{c}\right) \tag{B7}$$



We then have:

$$
i\bar{h}\big(k\hat{u}+l\hat{v}\big) = \frac{i\bar{h}}{\xi\nu}\Bigg( k\Bigg( \Big(3\eta l^2\tau_d + \nu ik\tau_d\cot\alpha + \frac{\nu\tau_d}{\bar{h}}\Big)\hat{s} + \Big(3\eta\bar{h}l^2 + \nu\Big)\gamma\bar{u}\hat{c}\Bigg)
$$

$$
+ l\Bigg( -3\eta kl\tau_d\hat{s} + \nu il\tau_d\cot\alpha\hat{s} - 3\eta\bar{h}kl\gamma\bar{u}\hat{c}\Bigg)\Bigg)
$$

$$
= \frac{i\bar{h}}{\xi\nu}\Bigg( k\Bigg( \Big(\nu ik\tau_d\cot\alpha + \frac{\nu\tau_d}{\bar{h}}\Big)\hat{s} + \nu\gamma\bar{u}\hat{c}\Bigg) + l\Big(\nu il\tau_d\cot\alpha\hat{s}\Big)\Bigg)
$$

$$
= \frac{i\bar{h}}{\xi}\Bigg( ik^2\tau_d\cot\alpha\hat{s} + \frac{k\tau_d}{\bar{h}}\hat{s} + k\gamma\bar{u}\hat{c} + il^2\tau_d\cot\alpha\hat{s}\Bigg)
$$

$$
= \frac{i}{\xi}\Bigg( i\bar{h}j^2\tau_d\cot\alpha\hat{s} + k\tau_d\hat{s} + k\bar{h}\gamma\bar{u}\hat{c}\Bigg)
$$

At steady state $\hat{s}_t = 0$, so we have the boundary condition:

$$
-ik\bar{u}\hat{s} = i\bar{h}(k\hat{u}+l\hat{v}).
$$

$$
= \frac{i}{\xi}\Bigg( ihj^2\tau_d\cot\alpha\hat{s} + k\tau_d\hat{s} + k\bar{h}\gamma\bar{u}\hat{c}\Bigg)
$$

$$
-ik\bar{h}\gamma\bar{u}\hat{c} = -hj^2\tau_d\cot\alpha\hat{s} + ik\tau_d\hat{s} + ik\bar{u}\xi\hat{s}
$$

$$
\frac{1}{\xi}\big(-ik\bar{h}\gamma\bar{u}\big)\hat{c} = \Bigg( -\frac{hj^2\tau_d\cot\alpha}{\xi} + ik\Big(\frac{\tau_d}{\xi} + \bar{u}\Big)\Bigg)\hat{s}
$$

$$
\frac{1}{\xi}\big(-ik\bar{h}\gamma\bar{u}\big)\hat{c} = \Bigg( -\frac{1}{t_r} + \frac{i}{t_p}\Bigg)\hat{s}
$$

$$
\big(-ik\bar{h}\gamma\bar{u}\big)\hat{c} = \xi p\hat{s}
$$

In agreement with Gudmundsson (2008) this leads to the steady-state transfer function:

$$
T_{sc}(k,l,t) = \frac{\hat{s}(k,l,t)}{\hat{c}(k,l)} = -\frac{ik\bar{h}\bar{u}\gamma}{p\xi} \tag{B8}
$$



Expanding the expression for $\hat{u}$ (Eq B6):

$$\hat{u} = \frac{1}{\xi\nu}\left(\left(3\eta l^2\tau_d + \nu ik\tau_d\cot\alpha + \frac{\nu\tau_d}{\bar{h}}\right)\hat{s} + \left(3\eta\bar{h}l^2 + \nu\right)\gamma\bar{u}\hat{c}\right)$$

$$= \frac{1}{\xi\nu}\left(\left(3\eta l^2\tau_d + \nu ik\tau_d\cot\alpha + \frac{\nu\tau_d}{\bar{h}}\right)\left(-\frac{ik\bar{h}\bar{u}\gamma}{p\xi}\right)\hat{c} + \left(3\eta\bar{h}l^2 + \nu\right)\gamma\bar{u}\hat{c}\right)$$

$$= \frac{\gamma\bar{u}}{\xi\nu p\xi}\left(\left(3\eta\bar{h}l^2\tau_d + \nu i\bar{h}k\tau_d\cot\alpha + \nu\tau_d\right)\left(-ik\right) + \left(3\eta\bar{h}l^2 + \nu\right)p\xi\right)\hat{c}$$

$$= \frac{\gamma\bar{u}}{\xi\nu p\xi}\left(\left(3\eta\bar{h}l^2\tau_d + \nu i\bar{h}k\tau_d\cot\alpha + \nu\tau_d\right)\left(-ik\right)\right.$$

$$\left. + \left(3\eta\bar{h}l^2 + \nu\right)\left(ik(\bar{u}\xi + \tau_d) - j^2\tau_d\bar{h}\cot\alpha\right)\right)\hat{c}$$

$$= \frac{\gamma\bar{u}}{\xi\nu p\xi}\left(\nu k^2\tau_d\cot\alpha\bar{h} + 3\eta\bar{h}l^2 ik\bar{u}\xi - 3\eta\bar{h}l^2 j^2\tau_d\bar{h}\cot\alpha + \nu ik\bar{u}\xi - \nu j^2\tau_d\bar{h}\cot\alpha\right)\hat{c}$$

$$= \frac{\gamma\bar{u}}{\xi\nu p\xi}\left(-\nu l^2\tau_d\cot\alpha\bar{h} - 3\eta\bar{h}l^2 j^2\tau_d\bar{h}\cot\alpha + 3\eta\bar{h}l^2 ik\bar{u}\xi + \nu ik\bar{u}\xi\right)\hat{c}$$

$$= \frac{\gamma\bar{u}}{\xi\nu p\xi}\left(-\left(3\eta\bar{h}j^2 + \nu\right)l^2\tau_d\cot\alpha\bar{h} + \left(3\eta\bar{h}l^2 + \nu\right)\left(ik\bar{u}\right)\xi\right)\hat{c}$$

$$\hat{u} = \frac{\gamma\bar{u}}{\xi\nu p}\left(-l^2\tau_d\cot\alpha\bar{h} + \left(3\eta\bar{h}l^2 + \nu\right)\left(ik\bar{u}\right)\right)\hat{c}$$

remembering that $\xi = 4\eta\bar{h}j^2 + \nu$, leads to the steady-state transfer function:

$$T_{uc}(k,l,t) = \frac{\hat{u}}{\hat{c}} = \frac{\gamma\bar{u}\left((3\eta\bar{h}l^2 + \nu)(ik\bar{u}) - l^2\tau_d\cot\alpha\bar{h}\right)}{\xi\nu p} \qquad (B9)$$

which is not as stated by Gudmundsson (2008).





Expanding the expression for $\hat{v}$ (Eq B7):

$$\hat{v} = \frac{1}{\xi\nu}\left(-3\eta kl\tau_d\hat{s} + \nu il\tau_d\cot\alpha\hat{s} - 3\eta\bar{h}kl\gamma\bar{u}\hat{c}\right)$$

$$= \frac{1}{\xi\nu}\left(-3\eta kl\tau_d\left(-\frac{ik\bar{h}\bar{u}\gamma}{p\xi}\right)\hat{c} + \nu il\tau_d\cot\alpha\left(-\frac{ik\bar{h}\bar{u}\gamma}{p\xi}\right)\hat{c} - 3\eta\bar{h}kl\gamma\bar{u}\hat{c}\right)$$

$$= \frac{kl\gamma\bar{u}\bar{h}}{\xi\nu p\xi}\left(3i\eta k\tau_d + \nu\tau_d\cot\alpha - 3\eta p\xi\right)\hat{c}$$

$$= \frac{kl\gamma\bar{u}\bar{h}}{\xi\nu p\xi}\left(3i\eta k\tau_d + \nu\tau_d\cot\alpha - 3\eta\left(ik(\bar{u}\xi + \tau_d) - j^2\tau_d\bar{h}\cot\alpha\right)\right)\hat{c}$$

$$= \frac{kl\gamma\bar{u}\bar{h}}{\xi\nu p\xi}\left(\nu\tau_d\cot\alpha - 3\eta ik\bar{u}\xi + 3\eta j^2\tau_d\bar{h}\cot\alpha\right)\hat{c}$$

$$= \frac{kl\gamma\bar{u}\bar{h}}{\xi\nu p\xi}\left((3\eta j^2\bar{h} + \nu)\tau_d\cot\alpha - 3i\eta\bar{u}k\xi\right)\hat{c}$$

$$\hat{v} = \frac{kl\gamma\bar{u}\bar{h}}{\xi\nu p}\left(\tau_d\cot\alpha - 3i\eta\bar{u}k\right)\hat{c}$$

remembering that $\xi = 4\eta\bar{h}j^2 + \nu$, leads to the steady-state transfer function:

$$T_{vc}(k,l,t) = \frac{\hat{v}}{\hat{c}} = \frac{kl\gamma\bar{u}\bar{h}(\tau_d\cot\alpha - 3i\eta\bar{u}k)}{\xi\nu p} \tag{B10}$$

which is also not as stated by Gudmundsson (2008).

## Appendix C: The inverse problem

The transfer functions ($T_{sb}$, $T_{ub}$, $T_{vb}$, $T_{sc}$, $T_{uc}$ and $T_{vc}$) describe the relationship between the Fourier transforms of bed topography ($\hat{b}$), bed slipperiness ($\hat{c}$), surface topography ($\hat{s}$) and surface velocity ($\hat{u}, \hat{v}$). If the bed topography and slipperiness are known then surface topography and velocity components are given by the forward model:

$$\hat{s} = T_{sb}\hat{b} + T_{sc}\hat{c} \tag{Eq. 26}$$

$$\hat{u} = T_{ub}\hat{b} + T_{uc}\hat{c} \tag{Eq. 27}$$

$$\hat{v} = T_{vb}\hat{b} + T_{vc}\hat{c} \tag{Eq. 28}$$

Non-dimensionalised this gives:

$$\hat{S} = T_{SB}\hat{B} + T_{SC}\hat{C} \tag{Eq. 26 non-dimensionalised}$$

$$\hat{U} = T_{UB}\hat{B} + T_{UC}\hat{C} \tag{Eq. 27 non-dimensionalised}$$

$$\hat{V} = T_{VB}\hat{B} + T_{VC}\hat{C} \tag{Eq. 28 non-dimensionalised}$$





Since the system is over-determined, we can use a weighted least squares inversion of equations 26, 27 and 28 to find the bed topography and slipperiness which are the most consistent with the ice surface. This is the same method used by Thorsteinsson et al. (2003) in their study of MacAyeal Ice Stream (formerly Ice Stream E), but is reproduced here in notation consistent with the rest of the equations we present.

In matrix form we have the forward model:

$$\mathbf{Y} = \mathbf{G}\mathbf{X} \tag{C1}$$

where $\mathbf{Y} = \begin{bmatrix} \hat{S} \\ \hat{U} \\ \hat{V} \end{bmatrix}$, $\mathbf{G} = \begin{bmatrix} T_{SB} & T_{SC} \\ T_{UB} & T_{UC} \\ T_{VB} & T_{VC} \end{bmatrix}$, $\mathbf{X} = \begin{bmatrix} \hat{B} \\ \hat{C} \end{bmatrix}$ and $\mathbf{E} = \begin{bmatrix} \sum_S^{-2} & 0 & 0 \\ 0 & \sum_U^{-2} & 0 \\ 0 & 0 & \sum_V^{-2} \end{bmatrix}$

A least squares inversion gives:

$$\mathbf{X} = \left(\mathbf{G}^H \mathbf{E} \mathbf{G}\right)^{-1} \mathbf{G}^H \mathbf{E} \mathbf{Y}$$

$$\begin{bmatrix} \hat{B} \\ \hat{C} \end{bmatrix} = \left( \begin{bmatrix} T_{SB}^* & T_{UB}^* & T_{VB}^* \\ T_{SC}^* & T_{UC}^* & T_{VC}^* \end{bmatrix} \begin{bmatrix} \sum_S^{-2} & 0 & 0 \\ 0 & \sum_U^{-2} & 0 \\ 0 & 0 & \sum_V^{-2} \end{bmatrix} \begin{bmatrix} T_{SB} & T_{SC} \\ T_{UB} & T_{UC} \\ T_{VB} & T_{VC} \end{bmatrix} \right)^{-1} \begin{bmatrix} T_{SB}^* & T_{UB}^* & T_{VB}^* \\ T_{SC}^* & T_{UC}^* & T_{VC}^* \end{bmatrix} \begin{bmatrix} \sum_S^{-2} & 0 & 0 \\ 0 & \sum_U^{-2} & 0 \\ 0 & 0 & \sum_V^{-2} \end{bmatrix} \begin{bmatrix} \hat{S} \\ \hat{U} \\ \hat{V} \end{bmatrix} \tag{C2}$$

where $^H$ is the Hermitian transpose and $^*$ is the complex conjugate.

For compactness we define the following:

$$K = T_{SB}^* T_{SC}/\sum_S^2 + T_{UB}^* T_{UC}/\sum_U^2 + T_{VB}^* T_{VC}/\sum_V^2$$
$$L = T_{SB}^* T_{SB}/\sum_S^2 + T_{UB}^* T_{UB}/\sum_U^2 + T_{VB}^* T_{VB}/\sum_V^2$$
$$M = T_{SC}^* T_{SC}/\sum_S^2 + T_{UC}^* T_{UC}/\sum_U^2 + T_{VC}^* T_{VC}/\sum_V^2$$
$$Y_b = \hat{S} T_{SB}^*/\sum_S^2 + \hat{U} T_{UB}^*/\sum_U^2 + \hat{V} T_{VB}^*/\sum_V^2$$
$$Y_c = \hat{S} T_{SC}^*/\sum_S^2 + \hat{U} T_{UC}^*/\sum_U^2 + \hat{V} T_{VC}^*/\sum_V^2$$

The least squares solution is then:

$$\begin{bmatrix} \hat{B} \\ \hat{C} \end{bmatrix} = \begin{bmatrix} L & K \\ K^* & M \end{bmatrix}^{-1} \begin{bmatrix} Y_b \\ Y_c \end{bmatrix}$$
$$= \frac{1}{LM - KK^*} \begin{bmatrix} M & -K \\ -K^* & L \end{bmatrix} \begin{bmatrix} Y_b \\ Y_c \end{bmatrix} \tag{C3}$$

This inversion is problematic where $LM - KK^*$ is small or zero, which is for small wavelengths (when k and l are large) or for topography or slipperiness perturbations which are aligned in the direction of ice flow (k = 0). Short wavelengths bed features and features aligned with ice flow are problematic because they cause flow disturbances in the ice which do not reach the surface in a measurable way. They can therefore not be inverted from the surface data.





To avoid this problem with an ill-conditioned inverse, Thorsteinsson et al. (2003) use a truncated version of $L'M' - KK^*$ as a filter, $F$ to remove problematic wavelengths. Their filter is

$$
F = \begin{cases} (LM - KK^*)/P, & \text{if } (LM - KK^*) \leq P \\ 1, & \text{if } (LM - KK^*) > P \end{cases}
$$

where

$$
P = \max\{\left(|LM - KK^*|\right) \bar{C}^{(p_{filt})}\}, \tag{C4}
$$

and $p_{filt} \leq 0$. This filter allows through all wavelengths where $LM - KK*$ is larger than P, but gradually filters out other smaller wavelengths. Smaller values of P give more detail, but may over-fit the surface data due to errors. Larger values of P under-fit the data and may leave out features actually represented by the data.

The filtered least squares solution is then:

$$
\begin{bmatrix} \hat{B} \\ \hat{C} \end{bmatrix} = \frac{F}{LM - KK^*} \begin{bmatrix} M & -K \\ -K^* & L \end{bmatrix} \begin{bmatrix} Y_b \\ Y_c \end{bmatrix} \tag{C5}
$$

*Author contributions.* RB, DG and AC developed the concept of the paper. DG advised on the use of the Gudmundsson 2008 equations; HO and DG derived the steady state transfer functions. AC advised on the inversion methodology. HO wrote the code, adapted the input data to the most useful form, and carried out the inversion. RB advised on the comparison to existing radar data. HO wrote the main body of the
text, but all authors contributed to the development of the final paper and associated figures.

*Competing interests.* The authors declare that they have no conflict of interest.

*Acknowledgements.* This work is an output of NERC E4 Doctoral Training Partnership, and the Glacial Habitat of Sub-glacial Thwaites (GHOST) project, a component of the International Thwaites Glacier Collaboration (ITGC).

*Financial support.* This work was supported by a Natural Environment Research Council (NERC) Doctoral Training Partnership grant
(NE/S007407/1), and by NERC and NSF grants as part of the International Thwaites Glacier collaboration (NSF PLR 1738934 , NERC NE/S006672/1, NERC NE/S006621/1, NERC NE/S006613/1 (GHOST), NE/S006796/1 (PROPHET)).



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
