# Peer review of "Inverting ice surface elevation and velocity for bed topography and slipperiness beneath Thwaites Glacier"

_The Cryosphere, 2021_

## Referee Comment (RC1)

**Review of Inverting ice surface elevation and velocity for bed topography andslipperiness beneath Thwaites Glacier by Ockenden et al.**

November 29, 2021

**1 Summary**

In "Inverting ice surface elevation and velocity for bed topography and slipperiness beneath Thwaites Glacier", Ockenden and co-authors present the application of a transfer function method for inferring basal topography and slipperiness from surface elevations to Thwaites Glacier in Antarctica. The authors claim that the method effectively captures the spatial pattern of variability in ice sheet topography as compared to radar flightlines. However, I am quite skeptical that the method is not being unduly influenced by the mean ice thickness that is being derived from existing thickness products. I am also skeptical of the utility of the product itself, given that it does not closely match observations, the mismatch is systematically biased, and the error estimate that accompanies it does not accurately reflect this mismatch. While I believe that the method is potentially interesting, for this manuscript to be suitable for publication, the authors need to 1) substantially increase the specificity of the description of their method, and 2) take a much more self-critical viewpoint of these results, in particular from the perspective of a reader that might wish to use the resulting product for some kind of downstream analysis.

**2 Line-by-line comments**

**L50** In what sense is linear perturbation theory underused?

**Eqs. 1/2** This form of the SSA implies some special coordinate system (flow-aligned, downhill in direction of flow). This needs to be justified to the reader.

**L65** Need to specify $u$ and $v$ as constant through depth by assumption.

**Eqs. 3-6** Some basic discussion of what assumptions are being made for linear perturbation analysis are necessary here. For example, the assumption

that the zero-order terms are spatially constant needs to be stated explicitly, otherwise it's confusing to see how these equations are derived.

**L73** Steady state means that the time derivative of the surface and bed elevations are zero. How is it justifiable to make that assumption for Thwaites?

**Eqs. 7-10** These need improved typesetting. It's not clear what is the argument of e.g. cos. Also, need to state that the Fourier representation of a variable is denoted by a hat, and that the hatted variables are the Fourier transform of the perturbations.

**L92** I think the expression 'variability in the Fourier components of the surface' is a bit opaque. This looks to me to just be the ratio of the fourier coefficients themselves as a function of wavenumber. Is the language about variability a reference to these being the fourier coefficients of the first-order perturbation?

**L93** While I recognize that this section is mostly a reproduction of Gudmondsson, it would be helpful to go a bit beyond just presenting these symbols and discuss just a bit what these mean, for readers that aren't already familiar with the antecedent work.

**L97** Should $h+\Delta s$ be $h+\Delta h$? They're the same in this case, but the difference should be mentioned for notational consistency.

**Section 2.1.3** Why is non-dimensionalization necessary here? it's not at all apparent that the transfer functions that appear in the supplement are simpler, and the inclusion of all this extra notation just makes the paper more confusing, especially given that there's no real motivation for why it's needed in the first place. Furthermore, the section itself is quite unclear. Where does $\bar{C}$ come from? The text here does a lot of hand-waving and should either be substantially expanded or eliminated.

**Eqs. 26–28** The arguments (k,l) to all of these terms needs to be retained in order to make clear that this function gives the amplitude of the surface perturbation for a given wavenumber as a function of the same for the bed. Also, I think that the traction transfer function subscripts should be capitalized?

**Sec 2.2/Appendix C** The inverse problem section is insufficiently described in these sections. Is this inversion being done for every wave number independently, or are they somehow coupled? Why is the 'filtering' method described a more sensible approach than a more common thing like a truncated eigendecomposition? How are the Fourier transforms of the observations performed? Most importantly, how are the error bars that appear in the later figures computed? This shows up qualitatively in the text, but it is not justified nor sufficiently detailed to be reproducible.

**Sec 2.3** Real topography does not have a delta function as its Fourier representation. It would be very helpful to see what the model's skill is for recovering topography that varies over multiple wavelengths simultaneously.

**L200** I think some care should be given to explaining a bit better why different 'patches' should give different results in overlapping regions. If I understand correctly, this is a direct result of the linearization and the fact that there's different assumed values of $\bar{h}$ and $\bar{u}$ being utilized in each. Is that true? Is there a way to communicate this clearly?

**L202** I'm confused by the mention of the SIA here: isn't this all based on the SSA?

**Sec 2.4** I'm concerned about the use of the BedMachine prior, particularly in the context of taking the mean of multiple overlapping blocks. It seems to me that if each of these overlapping blocks each has its own a priori mean being calculated from something else, and then these are being themselves aggregated, this is effectively injecting sub-50km scale a priori information. How do I know that what I'm looking at in Figs. 7 and 8 is not just a low-pass filter of BedMachine with some additional wiggles added from the inversion? This 'mean of means' would yield thickness results that 1) are biased too shallow and 2) are less skillful in regions of large bed slope, both of which are evident in the data.

**Sec 3.1** The error bounds reported are obviously unreliable. A 1-$\sigma$ credible interval ought to contain the data in 65% of instances. Simply looking at Figs. 7 and 8 show that this is not the case.

**L290** This statement regarding hyperparameters being sensible choices is not shown to be true in the text.

---

## Author Response (AR1)

Dear Editor, Reviewers and Cryosphere Discussion readers,

We would like to thank both reviewers for their very thorough and constructive reviews. In the following, we progress through both reviewers' comments (in bold) and our responses to them. If required we can provide a tracked-changes version of the revised paper, the previous version for reference to line numbers in reviewer comments, and a clean copy of the revised paper.

**Responses to Reviewer 1**

**1 Summary**

**In "Inverting ice surface elevation and velocity for bed topography and slipperiness beneath Thwaites Glacier", Ockenden and co-authors present the application of a transfer function method for inferring basal topography and slipperiness from surface elevations to Thwaites Glacier in Antarctica. The authors claim that the method effectively captures the spatial pattern of variability in ice sheet topography as compared to radar flight-lines. However, I am quite sceptical that the method is not being unduly influenced by the mean ice thickness that is being derived from existing thickness products.**

The discussion section of the reviewed submission (lines 311-322) explores the influence of the prior mean ice thickness (derived from Bedmachine Antarctica) on the results of the inversion. Only one average ice-thickness measurement is provided in each 50 by 50 km gridded region, which is minimal prior thickness information input compared to other methods, which interpolate between dense networks of radar grids and lines. The former Figure 9b showed the bed topography which would be produced if only the mean ice thickness input from Bedmachine was included, without the additional topographic variability derived from the inversion. This is contrasted with former Figure 9a showing the results of the inversion for the same region. We had considered that this demonstrated that all the topographic variability in the inverted bed topography is added by the inversion itself.

[Figure]

**Fig R1 (Above):** Bedmachine Ice Thickness, and a 50km gridded average (in each 50 by 50 km region a single value is used).

**Fig R2 (Below):** The new Figure 9 illustrating the negligible effect of the Bedmachine ice thickness on the results of the inversion

[Figure]

However, with this being raised as an issue in the review, we have reflected that the previous version lines 311-322 did not make the point emphatically enough. We have therefore re-run the inversion over the Lower Thwaites region (where we have existing swath radar), using a 50km gridded version of the Bedmachine Antarctica ice thickness (Figure R1) . In this alternate ice thickness dataset, each 50 by 50km region contains only one ice thickness value, which is the average over that region. The

average ice thickness in overlapping patches in this re-run therefore does not contain any more regionally specific values, which were of concern to the reviewer.

The results of this re-run (left plot of Figure R2/Figure 9 in new manuscript) with reduced prior information show the same short wavelength features as the original run with the full ice thickness dataset (right plot), illustrating that the method is not unduly influenced by the ice thickness derived from existing ice thickness products. We have replaced Figure 9 with these results, which we believe demonstrate this point much more clearly.

**I am also sceptical of the utility of the product itself, given that it does not closely match observations, the mismatch is systematically biased, and the error estimate that accompanies it does not accurately reflect this mismatch.**

We interpret that these comments largely refer to Section 2.4 of the paper, as they are raised in the in-line comments below.

We believe that a fairer way to describe the match with observations is that the inverted bed *does* match the observations extremely well in many places (such as the region around 105*W, 76.5*S which is shown in the right hand part of Figure 8) but not in others (such as Ridge Z, right hand part of Figure S2g). This reflects the validity of the physical assumptions made in this model (and other models) when inverting the surface topography for the bed conditions. Regions of mismatch are therefore still useful for identifying regions where interesting processes may be occurring, and which could be the sites of future field surveys.

We notice in particular that the method recovers shorter wavelengths of topography more accurately than longer wavelengths. We illustrate this in the manuscript by presenting the results of the inversion with the long wavelength (> 50km) Fourier components are removed.  (Figure R3/Figure 10 in the revised draft). The results then provide a significantly better match to the radar data. We have added the following text in the discussion section of the paper about these bandpass filtered results:

*In figures 7 and 8, it appears that the method estimates shorter wavelength topography more accurately than longer wavelengths. We demonstrate this in Figure 10, which shows results after wavelengths greater than 50km have been removed from all profiles. It is clear that the inversion identifies peaks and troughs in the bed, although the amplitude of these features is not always correct. Fourier components with a wavelength above 50km mainly represent the prior ice thickness information supplied to the inversion, as this is the large scale zero-order topography to which the first-order perturbations from the inversion are added. The greater match between the results of the inversion at the PASIN data after this bandpass filter therefore provides further evidence that the ice thickness derived from Bedmachine Antarctica does not influence the key results from the inversion.*

And we have also altered the text in the conclusions to reflect the fact that the best match to the data is seen at these shorter wavelengths:

*Comparison of the results of the inversion with radar grids and flight lines suggests that the inversion correctly identifies **short (< 50km horizontal) wavelength** features in the bed.*

[Figure]

**Figure R3:** The results of the inversion (orange) compared to PASIN radar flight lines (grey) and Bedmachine Antarctica (light blue) for an along flow profile and an across flow profile (locations shown in panels e) and f) respectively). Panels b) and d) show the bed profiles. Panels a) and c) show the results with any Fourier components between 40 and 50km in wavelength progressively damped with a half cosine filter, and any Fourier components over 50km in wavelength removed.

The standard deviation product which accompanies the bed topography is not an uncertainty estimate. It is a measure of the variability between the results from the various overlapping grids, which reflects how appropriate is the linearization of the equations for that region. For each grid, the topographic variations from the inversion are added to the average surface/bed slope. A higher standard deviation indicates that the linear assumptions are not appropriate in that region, for example if there is a change in the gradient of the bed across that region. We have added the following line to the end of Section 2.4 to make it clearer that this is not an error estimate, and therefore should not be expected to reflect the mismatch between this product and Bedmachine: *The standard deviation is **not** a measure of the error in the bed topography or bed slipperiness, and should not be interpreted as such.* A similar sentence has been added to the figure captions where

appropriate (ie Figure 5): *Standard deviations are a measure of variability between overlapped patches and should **not** be interpreted as a measure of the error in the bed topography or bed slipperiness.*

**While I believe that the method is potentially interesting, for this manuscript to be suitable for publication, the authors need to 1) substantially increase the specificity of the description of their method, and 2) take a much more self-critical viewpoint of these results, in particular from the perspective of a reader that might wish to use the resulting product for some kind of downstream analysis.**

The reviewer's in-line comments to which we respond below have provided useful guidance to enable us to increase the specificity of our description of the method.

We believe self-criticism of the results *was* evident in the reviewed submission, with our discussion (Section 4) reflecting on the problems with matching the amplitude of features in the inversion output to the radar observations. The location of topographic variability in the inversion results seems more robust, and particularly in the regions where swath radar measurements have been taken matches these observations well. Because of this perspective, we have purposefully not argued in this paper that the present product should be adopted across Thwaites Glacier rather than Bedmachine Antarctica;  but rather we view our method as another useful tool for exploring bed topography which is significant enough to influence the surface flow, and therefore ice-sheet behaviour.

**2 Line-by-line comments**

**L50 In what sense is linear perturbation theory underused?**

Our intention was to highlight the relative lack of application of linear perturbation theory to real ice-sheet data, building from its extensive development in theoretical publications. Ultimately however this is not an especially necessary point to try to insert into this sentence, so we have removed "under-used" from the sentence.

**Eqs. 1/2 This form of the SSA implies some special coordinate system (flow-aligned, downhill in direction of flow). This needs to be justified to the reader.**

We are using the system following Gudmundsson (2008). When introducing the equations, we have added a note that the coordinate system is flow-aligned in the direction of flow:

*Following Gudmundsson (2008), and working in a coordinate system tilted forward in the $x$ direction by the mean surface slope, α, we start with the shallow-ice-stream equations of motion (Macayeal, 1989):*

**L65 Need to specify u and v as constant through depth by assumption.**

We have changed the sentence to read:

*where* u and v *are the depth-independent velocity components in the* x, y *directions respectively,*

**Eqs. 3-6 Some basic discussion of what assumptions are being made for linear perturbation analysis are necessary here. For example, the assumption that the zero-order terms are spatially constant needs to be stated explicitly, otherwise it's confusing to see how these equations are derived.**

We have added a paragraph to the text before these equations explaining the assumptions required for linear perturbation analysis:

*Assuming that ice is a linear viscous medium (n = 1) and that there is a non-linear sliding law (m > 0), then the shallow-ice-stream equations can be linearised and solved analytically. We consider the spatial response to a small perturbation in basal topography, b, linearising around a reference model (¯h,¯s,¯b,¯u,¯v,¯c) with h =¯h+ Δh, s =¯s+ Δs, b =¯b+ Δb, u =¯u+ Δu, v = Δv, w = Δw and c =¯c. The zero order solutions are spatially constant, representing uniform flow down an inclined plane.*
*We, however, are interested in the first order momentum balance equations:*

**L73 Steady state means that the time derivative of the surface and bed elevations are zero. How is it justifiable to make that assumption for Thwaites?**

We explore the validity of the steady state assumption for Thwaites Glacier in the discussion section, and we have added the following line at this point to sign post the reader towards this discussion.

*Various points about the validity of the steady state assumption for Thwaites Glacier are raised in the discussion (Section 4).*

To briefly summarise the points in the discussion:

We do not have any repeat radar measurements from Thwaites Glacier so we have no idea how stable the bed is. However, observations at Pine Island glacier suggest that the bed there is not changing rapidly, and we expect the geology of the two regions to be similar. Changes in smaller features such as drumlins (like that observed at Rutford Ice Stream) would not affect this inversion anyway. The steady state assumption also applies to the ice surface, which has been observed to change. The largest changes are however close to the grounding line, but we don't cover the grounding line region in this inversion anyway due to the breakdown in ice physics there. Therefore, we consider the steady state assumption to be reasonable for the region which we consider.

**Eqs. 7-10 These need improved typesetting. It's not clear what is the argument of e.g. cos. Also, need to state that the Fourier representation of a variable is denoted by a hat, and that the hatted variables are the Fourier transform of the perturbations.**

We have changed the typesetting to improve the readability of the equations, and added a sentence to clarify that: *'Fourier transformed variables are denoted with a circumflex (ˆ).'*

**L92 I think the expression 'variability in the Fourier components of the surface' is a bit opaque. This looks to me to just be the ratio of the Fourier coefficients themselves as a function of wavenumber. Is the language about variability a reference to these being the Fourier coefficients of the first-order perturbation?**

We have clarified this expression to *'The transfer functions represent the ratio of the Fourier components of the surface to the Fourier components of the bed as a function of wavelength'*.

**L93 While I recognize that this section is mostly a reproduction of Gudmundsson, it would be helpful to go a bit beyond just presenting these symbols and discuss just a bit what these mean, for readers that aren't already familiar with the antecedent work.**

L93 at which this comment is raised is an introduction to symbolic abbreviations inherited from Gudmundsson (2008) that are used to simplify the algebra which is presented in the appendix.

On the general point of how much this paper should expand on the derivations already presented in Gudmundsson (2008) we appreciate the point but are attempting to strike a balance between

redundancy from repeating previously published work and this paper's overall length. We have largely retained the current presentation in this context, although we have made some clarifications in response to Reviewer 1's specific in-line comments above and below.

**L97 Should h+ Δs be h+ Δh? They're the same in this case, but the difference should be mentioned for notational consistency.**

When there is a change in the bed slipperiness then $\Delta h = \Delta s$, whereas if there is a change in the bed topography then $\Delta h$ is not necessarily $\Delta s$. We have changed this to clarify to read:
*We note that h = \bar{h} + Δh = \bar{h} + Δs.*

**Section 2.1.3 Why is non-dimensionalisation necessary here? it's not at all apparent that the transfer functions that appear in the supplement are simpler, and the inclusion of all this extra notation just makes the paper more confusing, especially given that there's no real motivation for why it's needed in the first place. Furthermore, the section itself is quite unclear. Where does ̄C come from? The text here does a lot of hand-waving and should either be substantially expanded or eliminated.**

We agree that the mathematical formulation of the non-dimensionalised transfer functions is not obviously any simpler than the dimensional versions. However, non-dimensionalisation allows use to use the equations to make more general statements about the behaviour of the system in terms of key variables, which helps us to understand the utility of the equations. For example, we use the ice thickness as the characteristic length scale, which allows us to say that the wavelength of landforms which can be resolved is equivalent to the ice thickness, a finding which is useful not just for Thwaites Glacier, but for applying these equations in other settings as well.

We have changed the wording in the text to reflect this motivation for non-dimensionalisation:
*These transfer functions can also be considered in a non-dimensional form, allowing us to make more general statements about the behaviour of the system in terms of key variables, such as ice thickness as the characteristic length scale.*

We have also added the following clarification to the text:
*The scale for slipperiness is given by ̄c/ ̄C, where ̄c is the mean dimensional slipperiness and ̄C is the mean non-dimensional slipperiness.*

**Eqs. 26–28 The arguments (k,l) to all of these terms needs to be retained in order to make clear that this function gives the amplitude of the surface perturbation for a given wavenumber as a function of the same for the bed. Also, I think that the traction transfer function subscripts should be capitalized?**

We have added the subscripts and the wavenumber arguments (k,l). We have also added in the sentence introducing these equations that they are functions of the wavenumbers k and l:
*The non-dimensional transfer functions (T_SB, T_UB, T_VB, T_SC, T_UC and T_VC) describe the relationship between the Fourier transforms of non-dimensionalised bed topography ( ˆB), bed slipperiness ( ˆC), surface topography ( ˆS) and surface velocity ( ˆU, ˆV ), as functions of the wavenumbers k and l.*

**Sec 2.2/Appendix C The inverse problem section is insufficiently described in these sections. Is this inversion being done for every wave number independently, or are they somehow coupled?**

We have extended the wording here so that it is clear that the inversion is done for each set of wavenumbers (k and l) independently:

*For each set of wavenumbers in Fourier space (k and l) we have three known variables ( ˆS(k,l), ˆU(k,l) and ˆV (k,l)) and two unknowns ( ˆB(k,l) and ˆC(k,l)), so the system is over-determined. We can therefore solve these equations independently for each wavenumber component of non-dimensional bed topography and slipperiness using a weighted least-squares inversion of equations 26, 27 and 28.*

**Why is the 'filtering' method described a more sensible approach than a more common thing like a truncated eigen-decomposition? ***

We believe the reviewer is referring to a truncated singular value decomposition approach to solving the least squares problem (Hansen 1987, https://link.springer.com/article/10.1007/BF01937276). We note that either approach involves inserting a priori information at small scales, due to the limitations of the shallow ice-shelf approximation itself, which limits expression of short length scales in bed and slipperiness. From this perspective both are likely equally sensible approaches, although a truncated eigen-decomposition of the normal equations may involve a costly eigen-decomposition of a large matrix, which our approach in Fourier space avoids.

**How are the Fourier transforms of the observations performed? ***

Fourier transforms are carried out using the standard 2 dimensional discrete fourier transform algorithm from the python numpy package. (https://numpy.org/doc/stable/reference/generated/numpy.fft.fft2.html#numpy.fft.fft2). This can be seen in the code which goes alongside this paper and which is available on both Github and Zenodo.

**Most importantly, how are the error bars that appear in the later figures computed? This shows up qualitatively in the text, but it is not justified nor sufficiently detailed to be reproducible. ***

The standard deviation product which accompanies the bed topography is not an error estimate, and we have changed the wording in the paper to make this clearer (see earlier response).

The basic inversion runs on a 50 km by 50km grid. We then run this on 9 (3 x 3 ) overlapping grids, which all cover the same central area, but in their top left, top centre, top right, middle left, centre, middle right, bottom left, bottom centre and bottom right respectively. See Figure 4 for a visual representation of this.

Within the overlapping region, we then have 9 values of the bed height at each geographical location. The standard deviation product is simply the standard deviation of these overlapping values.

$\sigma = \sqrt{\frac{\sum(x_i - \mu)^2}{N}}$ where σ is the standard deviation, $x_i$ are the bed heights 0 < i ≤ 9, μ is the mean bed height., and N is the number of observations (in this case 9).

This standard deviation is a measure of the variability between the various overlapping grids, which reflects how appropriate the linearization of the equations is for that region. A higher standard deviation suggests that the linear assumption is not appropriate in that region, for example if there is a change in the gradient of the bed across the region. However, we note that other factors such as the applicability of the steady state assumption and edge effects may also play a role in the standard deviation.

The code for this project (which shows exactly how the standard deviation is calculated using the standard formula given above), is available on both Github and Zenodo.

**Sec 2.3 Real topography does not have a delta function as its Fourier representation. It would be very helpful to see what the model's skill is for recovering topography that varies over multiple wavelengths simultaneously.**

Yes, we agree that real topography varies over multiple wavelengths simultaneously. However, because the inversion takes place in the spatial (Fourier) domain, it considers each wavelength separately. Sensitivity experiments carried out on topography varying over multiple wavelengths would therefore just be a summation of the sensitivity experiments carried out on each individual wavelength.

As we state on Line 175: *A two-dimensional Fourier transform decomposes an image into a weighted sum of two-dimensional sinusoidal basis functions. For this reason, all of our synthetic tests used sinusoidal bed topographies and slipperiness, as these are the most illustrative of the capabilities of the inversion.*

**L200 I think some care should be given to explaining a bit better why different 'patches' should give different results in overlapping regions. If I understand correctly, this is a direct result of the linearization and the fact that there's different assumed values of h̄ and ū being utilized in each. Is that true? Is there a way to communicate this clearly?**

Yes, this is correct. We have clarified the text here to read:
*For each grid point, we calculate 9 different ((but overlapping) inverted beds and then the mean bed topography and standard deviation. The standard deviation is not a measure of the error, but since the main approximation in the physics in the linearisation, we interpret the standard deviation to be a measure of non-linearity. In each of the overlapping grids, we use a set of zero order parameters (such as average ice thickness), and because these zero-order parameters vary between the grids, the linearisation is different, and the resulting beds are also different. However, inappropriate application of the shallow-ice-stream approximation, or edge effects could also be influencing this.*

**L202 I'm confused by the mention of the SIA here: isn't this all based on the SSA?**

We have made sure to use the term shallow-ice-stream approximation or shallow-ice-stream equations more consistently, and have also made the hyphenation more consistent.

**Sec 2.4 I'm concerned about the use of the BedMachine prior, particularly in the context of taking the mean of multiple overlapping blocks. It seems to me that if each of these overlapping blocks each has its own a priori mean being calculated from something else, and then these are being themselves aggregated, this is effectively injecting sub-50km scale a priori information.**

We have responded above to this in our first response to the reviewer's opening summary comments.

**How do I know that what I'm looking at in Figs. 7 and 8 is not just a low-pass filter of BedMachine with some additional wiggles added from the inversion? This 'mean of means' would yield thickness results that 1) are biased too shallow and 2) are less skillful in regions of large bed slope, both of which are evident in the data. ***

We have responded to this in our response to the reviewer's opening summary comments, and added a new figure which shows that this is not the case.

**Sec 3.1 The error bounds reported are obviously unreliable. A 1-σ credible interval ought to contain the data in 65% of instances. Simply looking at Figs. 7 and 8 show that this is not the case. (Reviewer 1)**

We have responded above to this in our second response to the reviewer's opening summary comments. Our specific response on the error bounds is in the third paragraph.

**L290 This statement regarding hyperparameters being sensible choices is not shown to be true in the text. (Reviewer 1)**

We have added two supplementary figures showing that the values p_filt = -2 and Σs = 0.001 are sensible for inverting the real data.

[Figure]

**Figure S3**. Bed topography results of the inversion for the Upper Thwaites region (see Figure 6 for location) with a variety of values of $p_{filt}$. Increasing p filters out increasing long wavelengths, and so a value of $p_{filt}$ = −2 is chosen to filter out noisy short wavelengths, while maintaining realistic bed features, as compared to Bedmachine Antarctica (Morlighem et al. 2019)

[Figure]

**Figure S4.** Bed topography results of the inversion for the Upper Thwaites region (see Figure 6 for location) with a variety of values of Σs. The weighting factor Σs controls the balance in the least squares inversion between the surface elevation and surface velocity data, with smaller values of Σs weighting the inversion towards the surface elevation. A value of Σs = 0.001 is chosen to produce realistic amplitude bed features, as compared to Bedmachine Antarctica (Morlighem et al., 2019)

**Responses to Reviewer 2**

**This study demonstrates a method for inverting for bed topography and basal slipperiness under ice sheets using more readily observable surface conditions (elevation and velocity), based on earlier theoretical work (e.g. Gudmundsson, 2008). Given the wealth of high-quality satellite-derived data that have become available in recent years, developing (and updating) tools and techniques that can extract even more information out of them is a useful endeavour. This manuscript demonstrates one such example, which, with the supporting code, could be of interest to a range of scientists working on ice sheets, from modellers to those designing targeted field campaigns. However, as it stands the Thwaites case study presented here is not overly convincing and it would be useful to see if the accuracy of the results can be improved when more prior information is included. I outline my major comments below, followed by a few minor points.**

**The technique presented here seems likely to be most successful in regions where we expect mass conservation to also be the most successful (i.e. fast flowing ice streams) and so the statement in the abstract on line 14 ("… a complimentary technique in regions where those techniques fail") isn't that convincing to me. Similarly, in the discussion, you state that mass conservation is more reliant on good-quality radar measurements (L322) compared to this technique, which I agree with, but I would also perhaps argue that in the regions where MC and this inversion technique perform the best, those radar measurements do tend to exist already owing to their glaciological significance, and in those cases MC perhaps might be the better choice?**

Thank you for your statements of qualified support for the methods that we have presented. We agree that there may well be some complementarity between regions of Antarctica for which both this method and mass conservation work well, the latter because sufficient measurements have already been acquired to support an effective application of mass conservation. It is not our intention to set our method up as an alternate solution to mass conservation or BedMachine, and to prevent this impression we have removed the following clause from the abstract (line 14) "…or as a complementary technique in regions where those techniques fail."

Our main intention with this paper is to demonstrate the potentially wide applicability of our technique, whether or not it is applied to regions that have already been well surveyed (although what is the definition of "well surveyed" is a moot point). In a significant part of Antarctica, including a significant part of the region presented here, the Bedmachine Antarctica bed is not produced using mass conservation algorithms, but using streamline diffusion. See Figure 3 in Morlighem (2020, https://doi.org/10.5067/E1QL9HFQ7A8M), who describe it as *'While not based on physics, streamline diffusion is a way to interpolate ice thickness between flight lines anisotropically.'* The inversion method presented here does not necessarily require fast ice flow, although if ice flow is too slow then the error in the ice velocities becomes more significant and so the method may be less sensitive. In addition, there are regions around Antarctica (such as Aurora Subglacial Basin, and Recovery Glacier) which have sparser radar coverage than Thwaites Glacier, where mass conservation techniques may not have enough prior information to pick up all the variability in bed topography which can be identified using this technique.

**Although I'm basing this assessment on the results in Figures 6-8 and it would be interesting to see what happens when the prior knowledge fed into the inversion improves. How easy would it be to incorporate thickness measurements from radar data into the inversion, as suggested in line 253? Using radar data directly, rather than aggregating the Bedmachine product, would help demonstrate the usefulness of this technique as an independent way of obtaining gridded ice thickness.**

As you note, this technique is not currently using the prior radar observations. This is partly due to the complications involved with including spatial prior information in the frequency domain, and because we wanted to avoid using a different less well known interpolation product or method. We plan to look further into this in future work, as we agree that it would be useful to include the original radar data rather than relying on the Bedmachine Ice thickness product.

**Following on from the point above, what happens when the average thickness grid resolution is reduced? Various other sensitivity tests have been rigorously carried out as part of the study, and so not including a test for the grid resolution seems like an obvious omission. What is the justification for choosing 50 km x 50 km? How easy would it be to incorporate thickness measurements from radar data into the inversion, as suggested in line 253? Using radar data**

**directly, rather than aggregating the Bedmachine product, would help demonstrate the usefulness of this technique as an independent way of obtaining gridded ice thickness.**

We have addressed the point about the resolution of the ice thickness grid in our response to reviewer 1's summary comments, and provided a new figure of this test which is now Figure 9 in the updated manuscript.

Sensitivity tests carried out on the size of the grid suggest that it is not particularly important. If it gets too small then the full wavelength of landforms is not included, and so some landforms are truncated. If it is too big then there are problems with going outside the ice catchment or including areas where the physical assumptions break down. 50 by 50 km seemed like a good compromise between these two limits for Thwaites Glacier.

As you note, this technique is not currently using the prior radar observations. This is partly due to the complications involved with including spatial prior information in the frequency domain, and because we wanted to avoid using a different less well known interpolation product or method. We plan to look further into this in future work, as we agree that it would be useful to include the original radar data rather than relying on the Bedmachine Ice thickness product.

**Do the results shown in Figure 1 suggest that in real world applications any unknown linear bedforms at an angle of less than 15 degrees to ice flow would not be resolved by this technique? This might be a very naïve interpretation, but is that why the Thwaites results are dominated by apparent bedforms that are approximately perpendicular to flow?**

Yes, we have interpreted the results of our sensitivity tests for the capabilities of the inversion to mean that linear bedforms with an angle of less than 15 degrees to ice flow would not be resolved. However, this still leaves 300 (360 – 4 x15 (either side of both 0 and 180)) degrees of variability for landforms, and does not necessarily mean that the landforms outputted by the inversion would be anywhere close to perpendicular to ice flow. The reason why this occurs in the Thwaites Glacier region is down to the underlying geology of the West Antarctic Rift System. Glacial landforms such as MSGL which are commonly aligned to ice flow and would be expected to overlie this geological control cannot be detected by this inversion technique anyway as they are smaller than the ice thickness in this region.

We believe the reviewer correctly inferred all of this and so we have not implemented any revisions in response to these queries.

**There is quite a lot of repetition between sections of the methods and the appendices, and equations in the main text and the appendix are referred to using the appendix reference (e.g. A7, A8 and A12 on line 87). I think the approach taken in Section 2.2 and Appendix C works the best – i.e. in the appendix starts with the relevant equations from the main text (using the same enumeration) and the further equations leading on from these (using the C1 etc numbering system). Could something similar be applied to appendix A and B?**

We have removed the duplication of derivation of equations from the appendix, and renumbered the equations so that all equations in the main text are numbered without reference to the appendix, and equations in the appendix are numbered with their corresponding numbers from the main text. This is the approach previously used in Appendix C.

**The discussion about the steady-state assumption (L271-282) is weighted towards justifying the assumption at the bed, whereas I would argue that in regions such as Thwaites the assumption is**

**more likely to break down at the surface due to flow acceleration and surface lowering. There are plenty of papers that could be cited here referring to observed changes in ice dynamics.**

We have extended the discussion of the steady-state assumption to consider the effects of ice surface lowering and acceleration as follows:

*The steady state assumption does not only apply to the bed but also to the ice surface. Ice surface lowering due to glacier thinning would also affect the steady state assumption, but since generally the ice surface lowers in a relatively uniform way, this would not have a significant effect on the first order variations in the ice surface, or the results of the model. More significant would be changes in the ice surface due to the filling and draining of subglacial lakes, but these changes are normally fairly localised, and would not propagate to the higher wavelength Fourier components. For Thwaites Glacier, the location of subglacial lakes is relatively well known (Smith et al., 2017; Hoffman et al., 2020; Malczyk et al., 2020), and we do predict troughs in these locations as expected. The ice surface also becomes more unstable closer to the grounding line, with increased crevassing which would affect the surface profile. However, since results in the region immediately adjacent to the grounding line are compromised by the different physics of the ice shelf anyway, this is not a significant concern. With these caveats, we therefore consider the steady-state assumption to be suitable for the purposes of this inversion.*

**Related to the steady-state assumption, how important is it to have temporally consistent observations, e.g. surface velocity and elevations from the same year or period? In data assimilation techniques used to initialise ice sheet models, non-contemporaneous input data can result in spurious signals in forward simulations. Would you expect to see non-physical artifacts in the bed topography and slipperiness, if the surface data are not consistent with one another?**

We note that the main aim of this work is not model initialisation, but to explore how much we can learn about bed topography from the ice surface observations. We agree though that in regions where the steady state assumption is not valid, then non-contemporaneous input data may influence the results, and we have added a paragraph on this point in the discussion of the paper:

*If the steady state assumption is valid, then the age of the datasets used in the inversion is not important. However, input data from different years or decades could also affect the steady-state assumption. The main surface expressions of known bed features appear to be fairly similar between REMA (Howat et al., 2019) (2008-2018), and the earlier Bamber DEM (Bamber et al., 2009) (2003-2008), supporting the validity of the steady-state assumption for Thwaites Glacier. However, we also note that non-steady-state changes in the ice surface may be the reason for some of the features we observe (such as Ridge Z, Figure 5) in the inversion output which are not seen in the airborne flight lines.*

**Minor comments:**

**L137-139 (Eqns 26-28) capitalise the subscripts in the last terms?**

This was also noted by Reviewer 1, and we have now changed this.

**Figure 1: Consider including an arrow to demonstrate the flow direction (or description in the caption). This would help the reader quickly interpret the angle to flow.**

We have added an arrow to show the flow direction.

**Line 203: "Shallow-ice" --> "Shallow-ice-stream". Elsewhere, hyphenation in this term is inconsistent.**

This point is addressed in our response to Reviewer 1 / Line 202 above.

**Figure 4: is this something that needs to be tested for each application? E.g. if the grid resolution changes?**

Because the equations are non-dimensionalised, the parameters can be considered in terms of the characteristic length scales. So if the ice thickness were to halve, then we would expect to be able to resolve landforms with half the wavelength. Changes in other parameters will have similar effects on the capabilities of the inversion which can be calculated without the need to run these experiments (although they can be done very quickly, and the code is available on Github/Zenodo).

If the grid resolution changes then there shouldn't be any changes in what can be resolved, because currently landforms less than the ice thicknesses (roughly 2km) are not well resolved, but the grid resolution is 120m. If the grid resolution were bigger than the ice thickness then this would be problematic, but satellite data are good enough that this is not a concern.

---

## Author Response (AR2)

We would like to thank the reviewer for their comments, which are addressed below.

**This paper has clearly been thoroughly reviewed and the original referees raised two major points**

**1. That the input of several overlapping 50 km means in ice thickness as prior information to a methods that aims to estimate finer scale variation in thickness results in a prior at effectively finer than 50 km resolution.**

**2. That the recovered ice thicknesses are notably lower than the radar derived thickness.**

**The authors answer point 1 by carrying out a new simulation, that as far as I understand it, computes a 50x50 km thickness map \*before\* deriving the thicknesses used to create the overlapping regions, which I think is reasonable but I had to re-read the new text a few times to work out what had been done, so I think a re-wording here may help**

We have updated the wording of the discussion about this new simulation to be clearer as suggested by the reviewer.

The original wording

To explore the role that the 50km averaged ice thickness plays in the results of the inversion, we re-ran the inversion over the Lower Thwaites region (where we have existing swath radar), using a 50km gridded version of the Bedmachine Antarctica ice thickness (Figure 9). In this alternate ice thickness input, each 50 by 50km region contains only one ice thickness value, which is the average over that region. The average ice thickness in overlapping patches in this re-run does therefore not contain any more regionally specific values, which may have been of concern.

The updated wording

To explore the role that the 50km averaged ice thickness plays in the results of the inversion, we computed a 50km gridded version of the Bedmachine Antarctica ice thickness (Figure 9) and then carried out a new inversion over the Lower Thwaites region (where there is existing swath radar). In this alternate ice thickness input, each 50 by 50km region contains only one ice thickness value, which is the average over that region.

**Point 2 is answered by noting that the method appears to obtain the medium wavelength variability (greater than a few ice thickness but less than 50 km) well, but the longer wavelength variations less well. This seems a reasonable response to me: the authors have made progress and documented its merits and flaws. I think the abstract needs a clear statement along these lines.**

We have altered the following sentence in the abstract to read:

'Although the topographic output from the inversion is less successful where the bed slopes steeply, it compares well with radar data from the central trunk of the glacier for medium wavelength features (5-50km).'